# Single-cell quantification and dose-response of cytosolic siRNA delivery

Hampus Hedlund ®[1], Hampus Du Rietz ®[1], Johanna M. Johansson ®[1], Hanna C. Eriksson ®[1], Wahed Zedan[1], Linfeng Huang ®[2], Jonas Wallin ®[3] & Anders Wittrup ®[1,4,5] ✉

Endosomal escape and subsequent cytosolic delivery of small interfering RNA (siRNA) therapeutics is believed to be highly inefficient. Since it has not been possible to quantify cytosolic amounts of delivered siRNA at therapeutic doses, determining delivery bottlenecks and total efficiency has been difficult. Here, we present a confocal microscopy-based method to quantify cytosolic delivery of fluorescently labeled siRNA during lipid-mediated delivery. This method enables detection and quantification of sub-nanomolar cytosolic siRNA release amounts from individual release events with measures of quantitation confidence for each event. Single-cell kinetics of siRNA-mediated knockdown in cells expressing destabilized eGFP unveiled a dose-response relationship with respect to knockdown induction, depth and duration in the range from several hundred to thousands of cytosolic siRNA molecules. Accurate quantification of cytosolic siRNA, and the establishment of the intracellular dose-response relationships, will aid the development and characterization of novel delivery strategies for nucleic acid therapeutics.

Small interfering RNA (siRNA) therapeutics are rapidly entering clinical use for multiple diseases. Lipid nanoparticle (LNP)-formulated siRNA targeting transthyretin (patisiran)[1] and three GalNAc-conjugated chemically stabilized free siRNA compounds (givosiran[2], lumasiran[3], and inclisiran[4]) have recently received clinical approval and several other substances are in clinical development. Both LNPs and GalNAc-conjugated siRNA target the liver, currently the organ most amenable to macromolecular delivery. A key impediment in efforts to improve siRNA delivery to other tissues has been a lack of tools to accurately detect and quantify successful cytosolic delivery of siRNA. Total tissue siRNA amounts do not directly correlate to biological effects due to the inefficiency and variability of cellular internalization and endosomal escape of the delivered siRNA[5]. For example, GalNAc-siRNA has been shown to be released from endo-lysosomal depots over several weeks[6] making cytosolic dose-response correlations difficult. Crucially, methods to quantify the cytosolic concentration of a delivered siRNA have been lacking[7] and the dose-response relationship

for cytosol-delivered siRNA has not been clear. As a consequence, it has generally not been possible to determine delivery efficiencies, and the scope for improvement, for different delivery strategies. Establishment of a generic dose–response relationship for a cytosol-delivered siRNA would be of great value for the study of various delivery strategies.

siRNA delivery mediated by transfection lipid has been shown to proceed by discrete release events resulting in a detectable cytosolic siRNA signal in live cells[8–10]. However, the detection level in previous experiments has not been low enough to capture varying degrees of knockdown, instead, all detected events resulted in maximal knockdown[10]. Other strategies to quantify absolute delivery amounts during lipid-mediated siRNA delivery have relied on either electron-microscopy of fixed cells[11] or ensemble measurements of Argonaute-2 (AGO2) immunoprecipitated siRNA as a surrogate marker for cytoplasmic siRNA[12,13]. Recently, quantitative endpoint analysis strategies to detect endosomal entry of drug delivery vehicles and peptides have

[1]Department of Clinical Sciences Lund, Oncology, Faculty of Medicine, Lund University, Lund, Sweden. [2]Wang-Cai Biochemistry Lab, Division of Natural and Applied Sciences, Duke Kunshan University, Kunshan, Jiangsu, China. [3]Department of Mathematical Statistics, Lund University, Lund, Sweden. [4]Skane University Hospital, Oncology, Lund, Sweden. [5]Wallenberg Center for Molecular Medicine, Lund, Sweden. ✉e-mail: anders.wittrup@med.lu.se

been presented[14–17]. However, with non-continuous methods, it is difficult to correlate single-cell delivery and subsequent knockdown kinetics. Continuous measurement strategies with initially non-fluorescent cargos that, upon interaction with proteins expressed in the cytosol, become fluorescent have also been devised[18,19]. Still, the limits of detection, linearity, and time-resolved information of such strategies are difficult to determine given the gradual conversion of the compounds to a fluorescent state. Finally, fluorescence correlation spectroscopy (FCS) and mass-spectrometry-based nanoSIMS have been used to quantify cytosolic delivery of both proteins[20] and anti-sense oligonucleotides[21,22]. While very promising, highly sensitive, and quantitative, these strategies have generally relied on the subjective selection of cytosolic regions of interest and are not easily amenable to high-throughput observer-independent quantifications. Intracellular dose-response determination of siRNAs has further been addressed using microinjection experiments in live cells with highly divergent results reported in the literature, suggesting cytosolic half-maximal inhibitory concentrations (IC50) or doses between 12 and several hundred siRNA molecules[23,24].

Here, we present a continuous imaging strategy in live cells to quantify endosomal escape events during lipid-mediated delivery with sufficiently few siRNA molecules to capture the dose-response interval with respect to knockdown outcomes in individual cells. Our method is based on array-confocal detection of fluorescent siRNA combined with

post-acquisition processing to exclusively measure cytosolic (non-vesicular) siRNA. The method enables detection and quantification of sub-nanomolar cytosolic siRNA release amounts from individual cytosolic release events and provides measures of quantitation confidence for each event. Single-cell kinetics of siRNA-mediated knockdown in cells expressing destabilized eGFP unveiled a dose–response relationship with respect to knockdown induction, depth, and duration in the range from a few hundred to several thousand cytosolic siRNA molecules.

## Results

### Detecting cytosolic siRNA during sub-nanomolar transfection

Our first aim was to improve the sensitivity and accuracy of cytosolic fluorescent siRNA detection compared to previous efforts[10]. Ultimately, we wanted to detect and quantify release events in cells incubated with siRNA doses resulting in sub-maximal knockdown (that is, doses around typical IC50 values) for relatively efficient siRNA sequences. To this end, we set out to visualize transfection lipid-mediated endosomal escape in cells incubated with sub-nanomolar concentrations of Alexa Fluor 647-labeled siRNA (AF647-siRNA). Using a GaAsP array-confocal detector (Airyscan, ZEISS)[25] frequent apparent endosomal escape events and cytoplasmic dispersions of AF647-siRNA were clearly visible during Lipofectamine 2000 transfection with 400 pM siRNA (Fig. 1a and Supplementary Movie 1), and also lower

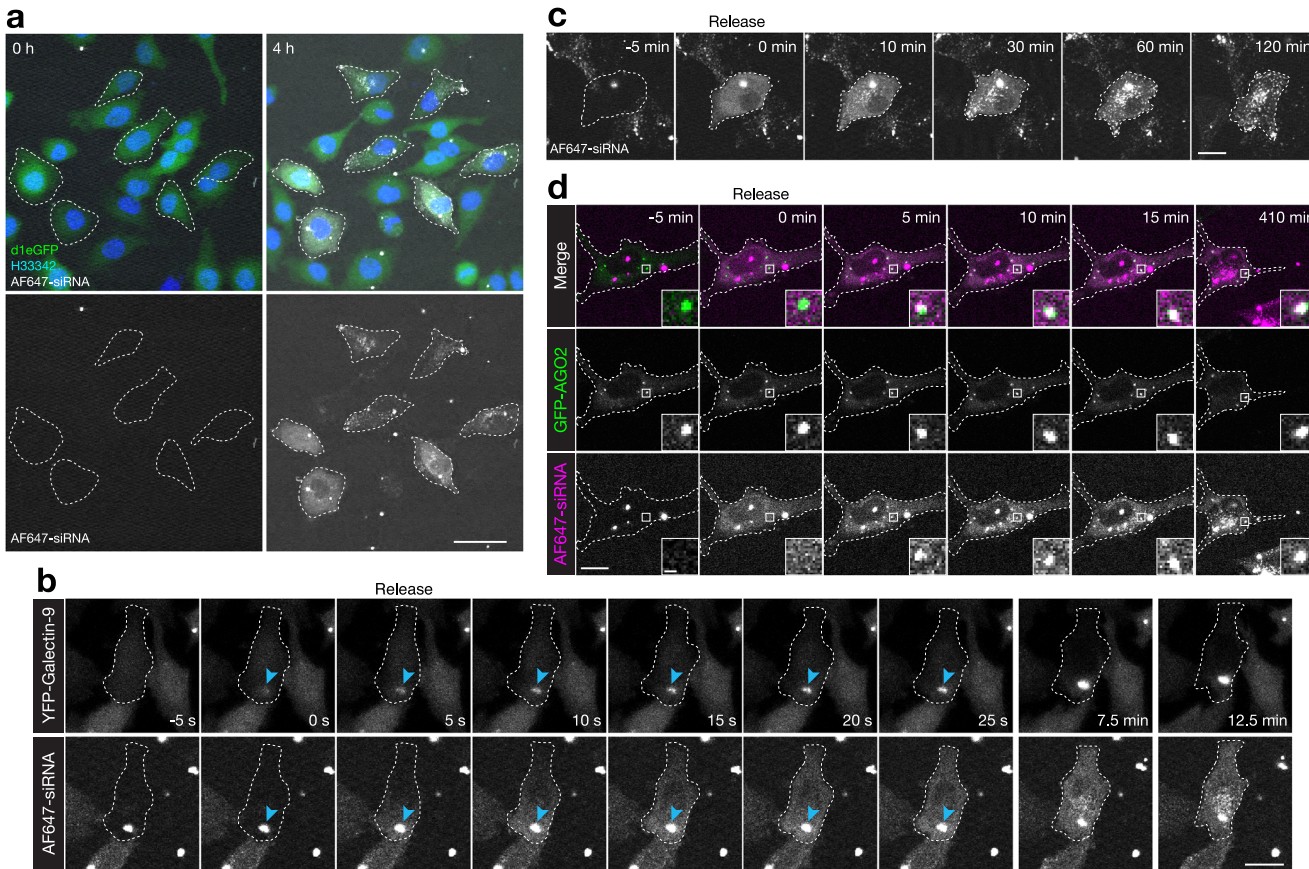

**Fig. 1 | Detecting cytosolic delivery of siRNA during sub-nanomolar siRNA transfection.** Airyscan confocal imaging of HeLa cells during treatment with lipoplexed AF647-siRNA. **a** HeLa cells stably expressing d1-eGFP exhibiting the cytosolic distribution of AF647-siRNA after 4 h siRNA treatment. The scale bar is 50 μm. Representative images of eight independent experiments. **b** HeLa cells expressing YFP-galectin-9 were imaged every 5 s during siRNA transfection, to visualize de novo galectin-9 recruitment to lipoplex-containing vesicles, followed by endosomal release and gradual cytosolic dispersion of AF647-siRNA. Outlines indicate cell boundaries and arrows indicate the releasing particle. The scale bar is

20 μm. Representative images of three independent experiments. **c** Representative images showing redistribution of AF647-siRNA to cytoplasmic foci after endosomal release and initial homogenous cytosolic dispersion. The scale bar is 20 μm. Images are representative of eight independent experiments. **d** The redistribution of siRNA-AF647(as) into cytoplasmic foci after release to the cytosol was monitored in cells expressing GFP-AGO2. The image detail shows a single AGO2⁺ structure. Scale bars, 20 μm; details, 2 μm. Representative images of two experiments. Cells were incubated with 0.4 nM (**a**, **c**) or 0.67 nM (**b**, **d**) siRNA.

doses. With this imaging, we could take advantage of the high dynamic range of the array-confocal detector, stemming from the use of multiple small GaAsP-detectors that tolerate a high photon flux (in total, distributed over all detectors) while maintaining a very low detection limit and high rejection of out-of-focus light. Thus, an Airyscan array-confocal detector can be used to detect cytoplasmic dispersion of siRNA over a large field-of-view (FOV) during low-dose transfection.

We next wanted to confirm that the sudden cytoplasmic dispersion of siRNA observed during imaging reflected the release of siRNA into the cytosolic compartment, where the RNAi machinery is located. Using the membrane damage sensor galectin-9, damages associated with the endosomal release of lipid-formulated siRNA can be detected[10]. Indeed, seconds after YFP-galectin-9 recruitment to AF647-siRNA lipoplexes, AF647-siRNA is released and gradually (over ~20 s) diffuses across the cytoplasmic space (Fig. 1b and Supplementary Movie 2). The dispersed siRNA is initially homogenous within the cytoplasm. However, 10–120 min after the release, the dispersed siRNA progressively accumulates in cytoplasmic foci of unknown identity (Fig. 1c and multiple examples in Supplementary Fig. 1), as has been observed previously[8]. Consistently, siRNA was excluded from the nucleus upon cytoplasmic dispersion during low-dose transfections. Thus, cytoplasmic siRNA dispersion is preceded by endosomal membrane damage (galectin-9 accumulation) and the released siRNA homogenously diffuses in the cytoplasmic space with an apparent diffusion speed consistent with free siRNA molecules and inconsistent with large supramolecular aggregates or membrane enclosed structures.

To further support the cytosolic localization and bioavailability of released siRNA, we treated HeLa cells expressing Argonate2-GFP (AGO2-GFP, a catalytic component in RNA-induced silencing complex, RISC) with siRNA-AF647(as) with the fluorophore on the active (antisense) strand of the siRNA. Immediately after cytosolic release, the siRNA was homogenously distributed, but within 5 min after the release, siRNA was seen to accumulate on pre-existing cytosolic AGO2-foci, as well as in AGO2-negative foci. Taken together, the fact that siRNA dispersion is concomitant with membrane damage, siRNA diffuses homogenously in the cytoplasmic space and within minutes can associate with pre-existing AGO2-foci, strongly support the notion that dispersed siRNA is located in the cytosol.

## Quantifying cytosolic siRNA signal in single cells

Fluorescence intensities can in principle be converted to absolute concentrations of a fluorescently labeled analyte using reference measurements of samples with known concentrations. For an FCS-calibrated point-scanning confocal, linear and quantitative fluorescence intensity readouts over several orders of magnitude, have been demonstrated[26]. Our imaging setup had precise and linear fluorescence readout for samples (fluid columns of ~10 μm) of solutions containing 1 nM to 1000 nM AF647-siRNA ($R^2 = 0.9932 \pm 0.0022$, mean ± s.d.) (Supplementary Fig. 2a, b). The Airyscan detector was furthermore shown to be highly linear also below 1 nM, using an approach where the AF647-siRNA concentration was gradually increased by the fluid exchange in a single, fixed imaging chamber, to minimize fluctuations induced by sample change and movement of the sample holder (Supplementary Fig. 2c). Acquiring images at maximal FOV (beyond recommended settings) was associated with notable vignetting (Supplementary Fig. 3), which was later corrected during post-acquisition processing. For accurate (low-noise) quantification, cytosolic siRNA fluorescence should ideally be measured in cross-sections of cells at the largest area possible, where the fluorescence intensity is homogenous (uniform cytosolic dispersion). Thus, fluorescence measurements in the cytosol of cells within minutes after release, offer a potential to quantify the cytosolic siRNA concentration down to 1 nM and below.

To exclusively measure the cytosolic fraction of the AF647-siRNA fluorescence, the contribution of non-cytosolic siRNA (non-released lipoplexes) must be removed. While highly fluorescent lipoplex particles present in the focal plane can be delineated and masked from images based on their high signal intensity, the hazy fluorescence contribution of out-of-focus particles are not as easily excluded. To solve this, we acquired two confocal $z$-planes of cells incubated with lipoplexes, spaced 4 μm apart (Fig. 2a, b). A cytosolic fluorescent protein signal (eGFP) was used to identify the boundaries of the cells in the lower plane ($z_1$). Bright, in-focus lipoplex particles present in either the lower ($z_1$) or upper plane ($z_2$) were identified and masked from both planes with an expanded margin in the AF647-siRNA channel. In this way, the contribution of out-of-focus fluorescence from lipoplexes in the upper part of the cells could be excluded from the lower plane image. Additionally, we measured and analyzed the median pixel intensity, which further suppressed any signal from localized fluorescence. Finally, cell nuclei were masked in the siRNA channel, to obtain a homogenous cytosol without fluorescence contribution from intact lipoplexes (Fig. 2c).

## Automated cytosolic release event detection

To facilitate unbiased analysis of cytosolic siRNA delivery in large numbers of cells, we designed an algorithm to automatically detect cytosolic release events in time-lapse imaging datasets. Using the imaging strategy described above and cell tracking (Supplementary Fig. 4), we monitored cellular siRNA fluorescence over time in individual cells, revealing sudden increases in the fluorescence intensity of variable magnitude during lipid-mediated transfection (Fig. 2d). Instances with abrupt siRNA signal increase were flagged as potential release events (Supplementary Fig. 5). The signal-to-noise ratio was improved by analyzing the entire cell including the nuclear region for release event detection, increasing the area of evaluation for a more robust event identification (Fig. 2e). Median filtering further decreased signal noise and made persistent intensity shifts more apparent (Fig. 2f). Thus, continuous monitoring of the cellular siRNA-fluorescence enables automated detection of sudden siRNA release events.

To evaluate the performance of the release event-calling algorithm and the sensitivity of the imaging setup, we used galectin-9 as an independent assay to detect siRNA release. Recruitment of YFP-galectin-9 to AF647-siRNA lipoplexes appears practically concurrently with endosomal escape at the frame rate (1 image every 5 min) used for long-term imaging (Fig. 3a). Consistently, in close to 200 analyzed release events, there was a clear increase in the cytosolic siRNA signal at the very moment that galectin-9 was recruited to a damaged vesicle (Fig. 3b and Supplementary Movie 3). Thus, the cytosolic siRNA signal increase is highly specific for membrane damage associated with endosomal release, demonstrating that basing a release event-calling algorithm on the increase in cytosolic siRNA signal is possible. Additionally, using galectin-9 as an independent method to detect release, we also verified that light exposure during lipoplex incubation did not induce release (Supplementary Fig. 6).

We next evaluated the specificity and sensitivity of the automatic event calling algorithm, with galectin-9 positive events as ground truth. We found that release events could be identified by the algorithm with 93% sensitivity and 56% specificity (Fig. 3c, d). In comparison, blinded manual inspection of time-lapse imaging datasets to identify sudden cytosolic dispersions of siRNA showed high agreement with events detected by galectin-9 recruitment (95% sensitivity, 97% specificity), but this approach was very labor intensive. By combining the automated identification of release events with subsequent manual inspection, a sensitivity, and specificity of 86% and 97%, respectively, were achieved. The combined automated detection and manual quality control provided a workflow capable of detecting

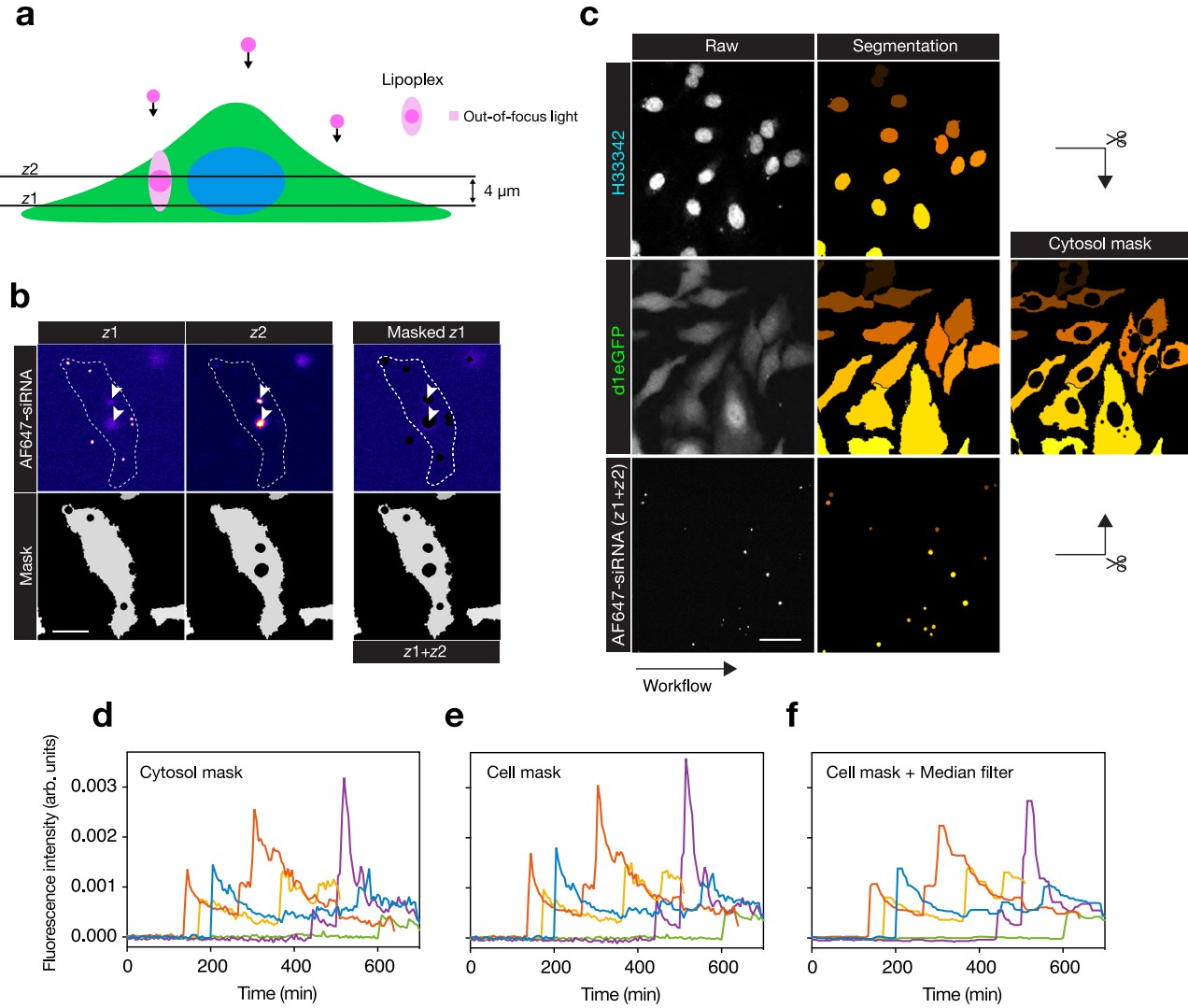

**Fig. 2 | Quantifying cytosolic siRNA signals in single cells. a** Two $z$-planes are acquired during live-cell microscopy. In-focus lipoplexes captured in the upper plane ($z2$) are used to mask out-of-focus light contaminating the lower $z$-plane ($z1$) used for cytosolic siRNA detection. **b** Cell (in $z1$) masked from bright lipoplexes, including out-of-focus light (in-focus in $z2$). The outline indicates cell and arrows indicate lipoplex fluorescence detected in both $z$-planes. The scale bar is 20 μm. **c** The final cytosol image mask used for siRNA detection after segmentation and removal of nuclei and lipoplexes. The scale bar is 50 μm. Images in **b**, **c** are representative of 23 independent experiments. **d**–**f** Examples of median AF647-siRNA fluorescence intensities measured in individual tracked cells (distinguished by color), **d** using a cytosol mask (nucleus and lipoplexes removed), **e** cell mask (lipoplexes removed), or **f** cell mask with a median filtering of the measurements using a 5-frame moving window. Cells were treated with 0.67 nM lipoplexed siRNA. Source data for **d**–**f** are presented in the Source Data file.

siRNA release with high specificity and sensitivity while keeping manual labor efforts reasonable.

We next sought to accurately quantify the cytosolic siRNA concentration after the release event. The typical cytosolic siRNA fluorescence intensity (median pixel) increases rapidly at the time of the release event and then gradually falls off (Fig. 2d), primarily because of cellular redistribution. The first frame with detectable cytosolic release most accurately corresponds to the released amount, because of the rapid, burst-like endosomal release and almost complete signal homogeneity within the cytosol. However, relying on a single frame for concentration measurements would result in noisy and potentially biased estimations. Therefore, we applied a mathematical model that captures the observed exponential decay of the cytosolic signal and fits the cytosolic signal intensities over 15 individual frames (75 min) to the model (Fig. 3e–g and Supplementary Fig. 7, for details, see Methods and Supplementary Note 1). This strategy enables estimations of absolute siRNA release amounts (expressed as a cytosolic concentration) for both low and high-magnitude release events. We designed the model to allow for two release events to occur within 14 frames (70 min), and to consider this as a single release event with a magnitude of the sum of the two individual events (Fig. 3g). Additionally, the model fit (coefficient of determination, $R^2$) provides a measure of accuracy and reliability of each individual cytosolic siRNA quantification. In summary, this acquisition and analysis strategy based on monitoring cytosolic siRNA fluorescence enables sensitive, specific, and quantitative siRNA release event detection.

## Cytosolic delivery is determined by the siRNA-to-lipid ratio

As a model system to determine the intracellular dose–response of siRNA we selected a modified HeLa cell line stably expressing destabilized eGFP (d1-eGFP)[27] with a short half-life of ~48 min (Supplementary Fig. 8), providing a rapid read-out of knockdown effects from the eGFP fluorescence intensity. Conventionally, siRNA sequence potency can be measured by lipid transfection to determine the extracellular IC50. To see if differences in extracellular IC50 between

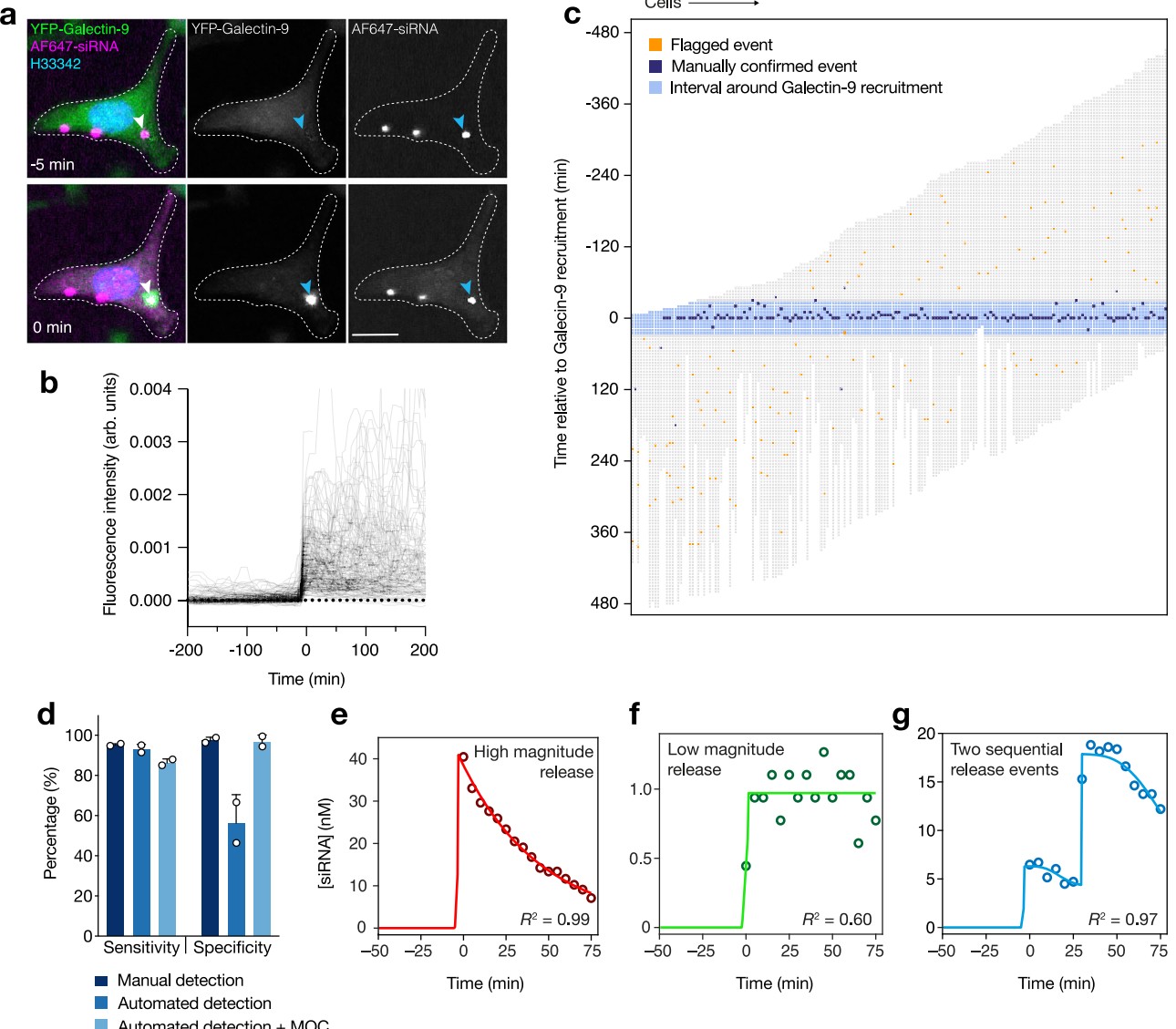

**Fig. 3 | Automated detection and absolute quantification of cytosolic siRNA during lipid-mediated delivery.** Airyscan confocal imaging of HeLa cells expressing YFP-galectin-9 every 5 min during treatment with lipoplexed AF647-siRNA. **a** Endosomal release and cytosolic dispersion of AF647-siRNA, coinciding with the recruitment of YFP-galectin-9 to the lipoplex-containing vesicle (indicated by arrowheads). The scale bar is 20 μm. **b** Measurements of cytosolic AF647-siRNA in single cells, aligned in time so that $t = 0$ is the first imaging timepoint with detectable de novo YFP-galectin-9 recruitment to lipoplex-containing vesicles. **c** Traces of individual cells (columns) with the recruitment of YFP-galectin-9 to vesicles containing siRNA-lipoplex, indicating the performance of the automated event detection in combination with manual quality control. Cell traces are aligned in time with $t = 0$ being the first frame with detectable galectin-9 recruitment. For **a–c**, $N = 187$ cells from two independent experiments. **d** Performance of manual and automated detection of endosomal siRNA release. Detection sensitivity and specificity were determined by comparing events indicated by galectin-9 recruitment and the manual or automated detection of AF647-siRNA release to the cytosol. Manual quality control (MQC) was performed after automated event detection, to exclude false positive events. Mean ± s.d. is shown. $N$ = two independent experiments. **e–g** Continuous monitoring of cytosolic AF647-siRNA fluorescence intensity was translated into absolute concentrations using condition-matched reference measurements. siRNA release magnitude estimations were made by fitting a mathematical model (lines) to the cytosolic siRNA concentration from single-cell measurements (circles). Examples are shown for the three different modeling approaches used, depending on the magnitude and kinetics of siRNA release: **e** a typical high-magnitude release event with exponential decay, **f** a high-noise low-magnitude release event (step-function), and **g** two separate events occurring in quick succession. Source data for **b–g** are presented in the Source Data file.

siRNA sequences are reflected in corresponding differences in the intracellular dose–response, we selected two siRNA sequences against eGFP with slightly different potency (relative IC50, compared to maximal knockdown, siGFP-1: 0.29 (CI 95%: 0.22–0.38) nM, siGFP-2: 0.65 (CI 95%: 0.52–0.81) nM; absolute IC50 siGFP-1: 0.34 (CI 95%: 0.27–0.43) nM, siGFP-2: 0.92 (CI 95%: 0.79–1.08) nM), as measured with flow cytometry using defined extracellular siRNA concentrations (Fig. 4a). Similar IC50 for siGFP-1 was also obtained on mRNA level, when measured with RT-qPCR (Supplementary Fig. 9).

We then set out to determine experimental conditions that would provide a wide spectrum of different cytosolic siRNA release amounts, to use as the basis for subsequent dose-response analysis. The siRNA release amounts were similar for the two sequences as well as for an inactive control sequence (targeting luciferase, siLuc) but were highly dependent on the ratio of siRNA to transfection lipids. Lipoplexes formulated with a higher siRNA to lipid ratio were larger (Supplementary Fig. 10) resulting in larger release amounts (Fig. 4b). Furthermore, the release amounts of each individual release event, at a

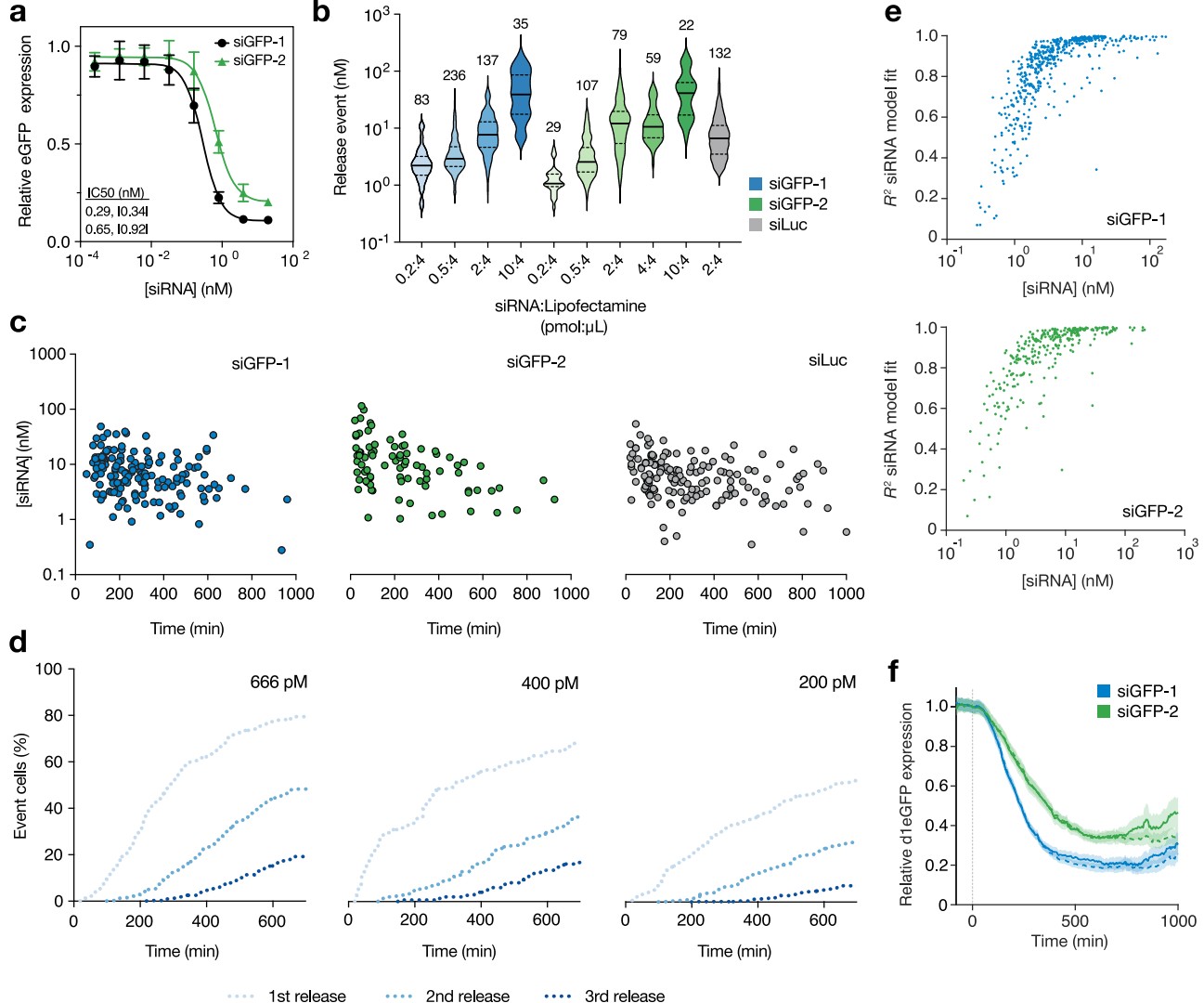

**Fig. 4 | Magnitude of cytosolic delivery is determined by the siRNA-to-lipid ratio.** HeLa cells stably expressing d1-eGFP were treated with lipoplexed AF647-siRNA targeting eGFP (siGFP-1 or siGFP-2) and analyzed by **a** flow cytometry or **b**–**f** live-cell microscopy followed by single-cell analysis. **a** d1-eGFP knockdown evaluated after 24 h siRNA-lipoplex incubation. Mean ± s.d. is shown. Relative and |absolute| IC50-values are indicated. N = 3 (siGFP-1) and 4 (siGFP-2) independent experiments. **b** Magnitudes of siRNA releases using lipoplexes formulated with different amounts siGFP-1 or siGFP-2 (constant for siLuc). The solid line is the median and the dashed lines are the 25th and 75th percentiles. N = cells indicated above violins. **c** Magnitude and time of siRNA release events during incubation with 200–666 pM lipoplexed AF647-siRNA. Lipoplexes were formed with a constant concentration and siRNA:lipid ratio (2:4 pmol:μl) with different volumes added to cells. The start of the experiment is t = 0. Single-cell measurements are shown,

N = 151, 84, and 143 cells from three, one, and four independent experiments, respectively. **d** Cumulative frequency of siRNA release events in cells incubated with indicated concentrations of lipoplexed siGFP-1, siGFP-2 or siLuc. Mean values are shown. N = 2, 1, and 5 independent experiments, respectively. **e** Peak cytosolic siRNA concentrations estimated by release modeling and corresponding goodness-of-fit statistics ($R^2$). Individual release events are shown. N = 546 and 316 events from 13 and 6 independent experiments, for siGFP-1 and siGFP-2, respectively. **f** d1-eGFP knockdown following one (solid lines) or multiple (dashed lines) siRNA release events per cell. Typically, after a second event is detected, subsequent measurements are excluded from analysis (solid lines). The first event occurs at t = 0. The line is the mean, shaded area is an 80% confidence interval. N = 546 and 316 cells from 13 and 6 independent experiments for siGFP-1 and siGFP-2, respectively. Source data are presented in the Source Data file.

given ratio between lipid and siRNA, varied over more than an order of magnitude, reflecting the heterogenous size of individual lipoplexes. The release magnitude was relatively independent of the time since the addition of lipoplexes to the cells (Fig. 4c). Adding more lipoplexes to the cells (that is, adding a higher dose siRNA and a correspondingly higher dose transfection lipid at a constant ratio) resulted in more release events, occurring more rapidly (Fig. 4d) but with small effects on the average release magnitude (Supplementary Fig. 11). Importantly, very frequent events resulted in shorter traces of individual cells (in time) from the first to the second event. To obtain the intracellular dose–response relationship from a single intracellular siRNA dose, masking or disregarding expression data after a

second event is necessary as this data would otherwise confound the dose-response determination. Thus, experimental conditions were optimized to achieve long analyzable traces before confounding second-release events.

Based on the observations above, we collated multiple experiments with varying lipoplex doses and siRNA-to-lipid ratios for the two siGFP sequences incubated with HeLa-d1-eGFP cells. We obtained a wide variety of release magnitudes with the quantification model fit ($R^2$) typically above 0.75 for release magnitudes of 1 nM or more (Fig. 4e). This high degree of model fit results in low estimated uncertainties. Taking the main sources of measurement uncertainty into account (including imprecision in reference curve calibration and

bleaching-induced variability) the error for individual release quantifications was estimated to <20% (relative std. dev.) for the vast majority of release events (Supplementary Note 2, Supplementary Fig. 12).

To monitor the effects on eGFP expression, the eGFP fluorescence intensity of tracked cells was corrected for bleaching, mitosis-induced fluctuations, and experiment-specific factors as described in Methods (Supplementary Figs. 13 and 14, and Supplementary Note 3). When averaging the eGFP expression of each individual cell experiencing a release event in this collection of data (with the time of the first release event set to 0 for each cell), knockdown was more prominent with the more potent siGFP-1 sequence compared to siGFP-2 (Fig. 4f). In addition, when comparing the eGFP expression of cells with or without masking of data after a second release event, a longer knockdown duration can be appreciated when cells having had a second release event are included (Fig. 4f). These secondary events happen at later timepoints during the monitoring, limiting the contribution to knockdown to primarily the end of the observation. In summary, evaluating eGFP expression after the endosomal escape of variable amounts of siRNA (up until either a second release event or the cell is lost in tracking) can provide quantitative measures of knockdown kinetics for at least up to 17–20 h. A schematic of the key components of the analysis pipeline is shown in Supplementary Fig. 15.

### Knockdown kinetics is dependent on cytosolic siRNA amounts

We then turned to evaluate the dose-dependency of the eGFP knockdown. For this analysis, the cells were grouped into five quintiles depending on the model-based magnitude of siRNA release for both siGFP sequences, excluding very low-confidence siRNA quantifications ($R^2 < 0.3$). The maximal cytosolic siRNA fluorescence intensities after the release were similar to the model estimates of siRNA release magnitude within each quintile, but the maximal fluorescence intensity was consistently lower at high release magnitudes (Supplementary Fig. 16a, b). Cells exhibiting two closely spaced release events were accurately quantified by the model while the maximal magnitude of fluorescence intensity underestimated the total release amount, highlighting the advantage of the model-based quantifications (see an example of this effect in Fig. 3g).

An initial siRNA-knockdown dose dependency could then be obtained by comparing the resulting knockdown for each quintile to the siRNA release amount of the quintile. For siGFP-2, eGFP knockdown varied between 38% and 84% at 10 h after release of 0.93 (median, interquartile range: 0.68–1.24) nM and 37.2 (22.2–55.8) nM (Fig. 5a and Supplementary Fig. 16). Given the typical cytosolic volume of HeLa cells used in this study of 5000 fL (interquartile range: 3800–5900 fL, Supplementary Fig. 17), the lowest quintile corresponded to the release of 2800 molecules (interquartile range: 2000–3700 molecules in an average sized cell) and the highest quintile corresponded to the release of 110000 molecules (interquartile range: 67,000–170,000 molecules). For the more potent siGFP-1 sequence knockdown was 72% after 10 h at both 1.21 (1.07–1.29) nM (3600 molecules, 3200–3900 i.q. range) and 2.05 (1.97–2.23) nM (6200 molecules, 5900–6700 i.q range). However, the accuracy of the siRNA quantifications was substantially worse, particularly in the lowest quintile (mean $R^2$, quintile 1: 0.62, quintile 2: 0.80). Restricting the analysis to only highly reliable quantifications ($R^2 > 0.75$) yielded a clear dose–response relationship for both sequences with respect to knockdown induction kinetics, knockdown depth (nadir) and knockdown duration (Fig. 5b). Thus, this combined strategy of monitoring siRNA delivery and target knockdown in live-cells with a model-based analysis can be used to elucidate the dose–response relationship of potent siRNA sequences.

We next wanted to determine the intracellular IC50 values for both siRNA sequences. As the degree of knockdown is dependent on time since release (Supplementary Movie 4), absolute intracellular IC50 values (that is, the cytosolic concentration that results in 50% knockdown) is highly dependent on the time point chosen. Relative IC50 values (the concentration with half-maximal inhibition at that specific time point) are potentially more stable. Plotting the eGFP expression level of single cells relative to the siRNA release amounts at various time points revealed comparatively stable relative IC50 values, varying between 0.60 (CI 80%: 0.34–0.96) and 0.37 (CI 80%: 0.15–0.77) nM for siGFP-1 and 3.02 (CI 80%: 2.01–4.80)–1.60 (CI 80%: 0.96–2.74) nM for the less potent siGFP-2 at 6 h and 10 h, respectively (Fig. 5c–e). Thus, the relative intracellular IC50 of a siRNA sequence is a measure of its potency and it correlates to conventionally measured extracellular IC50.

We finally wanted to obtain an intuitively understandable intracellular siRNA concentration or number of molecules that results in 50% knockdown for the two sequences respectively, that is, absolute IC50 values at the time point of a maximal knockdown. For this, we observed that 10 h after release, knockdown was close to the nadir or at a plateau for all release magnitudes for both sequences (Fig. 5b). This was also evident in the pooled data with release events of all magnitudes (Fig. 4f). At 10 h after release, 50% knockdown (absolute IC50) was determined to be at 0.31 (CI 95%: 0.18–0.48) nM (970 molecules, CI 95%: 540–1400) and 2.29 (CI 95%: 1.44–3.59) nM (6900 molecules, CI 95%: 4300–11,000) respectively for siGFP-1 and siGFP-2. However, relying solely on expression measurements at single time points makes the estimation potentially overly dependent on the specific measurements at this time point. To gauge the robustness of the IC50 determinations, we, therefore, adjusted a mathematical model to the dose-dependent knockdown kinetics, taking advantage of expression measurements from all time points and all release magnitudes (Fig. 5f, see Supplementary Note 1). Based on this model, absolute IC50 at 10 h was estimated to be 0.36 (0.14–0.65) nM for siGFP-1 and 3.36 (2.15–4.87) nM for siGFP-2 (median and 95% CI, calculated by bootstrapping), analogous to the values determined solely at 10 h. Therefore, we judge the experimentally derived absolute IC50 values determined at 10 h to be robust estimates for absolute intracellular IC50 values.

## Discussion

Here, we present a method to measure absolute cytosolic siRNA delivery amounts during lipid-mediated delivery. This method enabled us to determine single-cell intracellular dose-response relationships for siRNA knockdown of a reporter gene, including IC50, knockdown nadir, and duration. We show that half-maximal knockdown induction and knockdown nadir are reached at a few hundred picomolar (~1000 intracellular molecules) with a potent siRNA sequence (siGFP-1). We also show that higher siRNA doses, beyond initial knockdown saturation, prolong knockdown. The duration differs between cells having received ~13 and ~37 nM (~40,000 or ~110,000 molecules) of a less potent sequence (siGFP-2), highlighting a very large dynamic range in knockdown responses.

Previous cytosolic siRNA dose–response estimates have provided data at single time points without information on cell-to-cell variability[28]. Cytosolic IC50 was estimated to be ~2000–4000 molecules using electron microscopy of gold-labeled siRNA[11] and through AGO2-immuno precipitation, IC50 was determined to be 10–110 RISC-loaded siRNA molecules (but with unclear RISC-loading efficiencies)[12]. As a comparison, using less accurate (compared to this study) cytosolic concentration measurements, an estimated ~1.6 nM of siGFP-1 resulted in close to maximal knockdown[10], similar to the almost maximal knockdown induction seen above ~3 nM here. While our initial aim was to establish the dose–response and to measure the intracellular IC50, our knockdown kinetics analysis revealed that this is a dynamic concept—the IC50 values are strongly dependent on time since siRNA delivery (in addition to sequence potency). The nadir in eGFP expression is at 10–15 h after release for lower doses but this timing is dose-dependent and for very high siRNA releases, nadir does not seem

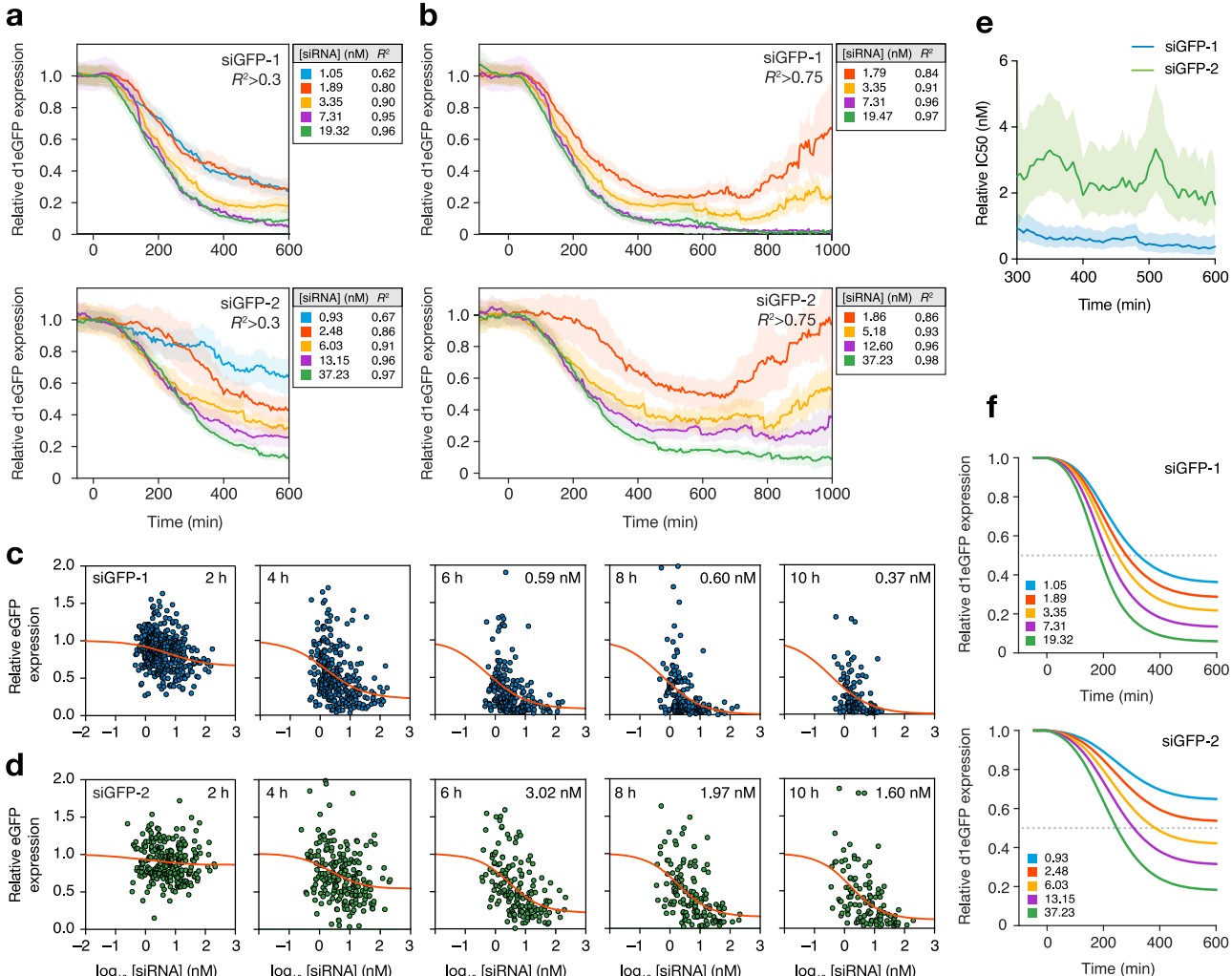

**Fig. 5 | Single-cell knockdown kinetics are dose-dependent with respect to cytosolic siRNA amounts.** HeLa cells stably expressing d1-eGFP were treated with 40–2000 pM lipoplexed AF647-siRNA targeting eGFP. A confocal microscope with an Airyscan detector was used for live-cell imaging, followed by single-cell analysis. **a**, **b** Cells were ordered and divided into equal groups based on the model-estimated magnitude of siRNA release events. Traces are aligned with $t = 0$ at the time of cytosolic siRNA detection. Lines are mean eGFP expression per quantile, shaded areas are 80% confidence intervals. Median cytosolic siRNA concentration (nM) and mean $R^2$-value per quantile are indicated. Model-fit ($R^2$) thresholds of >0.3 (**a**) and >0.75 (**b**) were used. $N \geq 92$, 58 ($R^2 > 0.3$) 90, and 59 ($R^2 > 0.75$) cells per quantile for siGFP-1 and siGFP-2, respectively. **c**, **d** Single-cell knockdown kinetics between 2 and 10 h after siRNA release. Model-estimated peak cytosolic siRNA

concentration and eGFP expression relative to $t = 0$ is shown per cell, for **c** siGFP-1 and **d** siGFP-2. Release events with model $R^2 > 0.3$ are shown. Red line is the sigmoidal curve fitted by least squares regression. Absolute knockdown IC50 (nM) is indicated. Between 2 and 10 h, $N$ ranges from 419 to 119 and 264 to 117 cells, for siGFP-1 and siGFP-2, respectively. **e** Relative knockdown IC50 for each time point based on data shown in (**c**, **d**). Time from the detected release event is indicated. Cells with release events with model $R^2 > 0.3$ are shown. The line is mean IC50 and the shaded area is 80% confidence intervals. **f** A model was used to estimate eGFP knockdown mediated by siGFP-1 or siGFP-2, predicted from the median peak cytosolic siRNA concentration after release in each quantile (indicated in graphs). All data shown are from 13 and 6 independent experiments, for siGFP-1 and siGFP-2, respectively. Source data are presented in the Source Data file.

to be reached at 20 h. The mechanism for this slow induction of full knockdown at high doses is not clear but could involve a gradual increase in siGFP-associated RISCs when existing miRNA-occupied RISCs continuously are replaced with newly synthesized RISCs. "Free" siRNA molecules could thus act as a cytoplasmic depot of siRNA[13], distinct from a recently demonstrated long-term non-cytoplasmic depot during GalNAc-siRNA delivery[6].

An important aspect of our detection method is the ability to estimate the reliability of each individual cytosolic siRNA quantification. We show that low model fit (due to noisy low-intensity measurements) is the primary driver of siRNA quantification uncertainty at small release magnitudes, while systematic error sources of ~10% dominate at large release magnitudes. The vast majority of release quantifications have estimated errors of <20% supporting the notion that single-cell data analysis is possible and appropriate. The reliability

of the quantifications can thus be appreciated by both a high degree of model fit at estimated release amounts above 0.5–1 nM (with low estimated errors in the siRNA quantification) and ultimately by the clear dose–response in knockdown for release events with good model fit (high $R^2$).

The Airyscan detector used here is optimized for imaging within a small area in the center of the FOV of the confocal system, and only within this area, the alignment is optimal for super-resolution imaging. In our experiments, we have used the full FOV of the microscopy system but without extracting any super-resolution information, essentially using the detector as a single, high dynamic range confocal detector with a 2.0 Airy unit pinhole. Other detection strategies are conceivable including modern, high quantum-yield sCMOS-based, spinning-disk confocal systems with sparse pinhole patterns and high out-of-focus light suppression. Various implementations of light sheet

microscopy, for example, lattice light sheet[29], HILO[30], or other single-objective light sheet designs[31] would be attractive from a detection limit perspective but would generally restrict FOV and experimental throughput.

In the present study, the siRNA cargo is delivered with a cationic transfection lipid, resulting in larger discrete delivery events than what is achieved with clinical-grade LNPs or siRNA-ligand conjugates. The large discrete events and the easily masked intact lipoplexes make the strategy presented here possible. It is important to note that the strategy is not directly translatable to smaller delivery vehicles or particles, for which it would be more difficult to mask and remove non-released particles from subsequent cytosolic quantifications. We have not determined a firm size limit, but particles smaller than a few hundred nm would probably be too faint and generate a too-small cytosolic signal to be directly analyzable with the current strategy. Detecting dispersed siRNA in the cytosol after a single LNP-delivery event would conceivably require a single-molecule detection strategy. This would be highly challenging given the substantial fluorescence from non-released particles, but current rapid improvements in fluorescence microscopy will be directly applicable to such efforts. How multiple small release events occurring over time (such as from LNPs), in aggregate contribute to knockdown kinetics and potency is currently unclear, but the modeling strategies presented here offer a potential avenue to address this question. An additional advantage of the presented strategy is the possibility to combine it with other fluorescence microscopy readouts, as demonstrated here with parallel expression level monitoring or AGO2 colocalization.

Our method measures and estimates an apparent cytoplasmic concentration. The absolute concentrations are derived by comparison to a reference curve, similar to what is done in conventional absolute qPCR or other absolute quantitative methods. However, the concentration is measured over the space containing both the cytosolic fluid volume and all vesicles and other membrane-enclosed organelles within the cytoplasmic space. The effective siRNA concentration in this smaller volume (that is, only the cytosolic fluid volume) is higher than the estimated concentration over the whole cytoplasmic space. The estimated number of released molecules is unaffected by this distinction. From a quantitative perspective, it is important to note that the siRNA diffuses homogenously directly after the release event, with no hints of supramolecular aggregates in the signal distribution or the diffusion speed. The measured intracellular concentrations are also orders of magnitude below the levels where fluorescence quenching effects interfere with quantitative reliability and linearity.

This study is limited to determining the dose-response relationship of two siRNA sequences and a destabilized eGFP reporter gene in a single cell line. Conventionally determined extracellular IC50 is a poorly defined concept as it will vary depending on the internalization efficiency of the cell line and lipid used, volume, and geometry of the sample well, because of the particulate nature of lipoplexes that sediment onto cells. The intracellular dose–response presented here is not plagued by the same limitations. However, it is noteworthy that for both sequences, the classically determined extracellular IC50 values are numerically close to the intracellular IC50 values determined here, though this is a consequence of the specific internalization efficiency of the HeLa cell line used. Future studies will elaborate on how the intracellular dose–response varies between different cells, tissues, target genes, and siRNA sequences. Indeed, recently it was suggested there is cell type-specific variability in mRNA susceptibility and siRNA efficiency[32], and mitotic activity[33], target mRNA abundance[34], and AGO2-expression[35] are known to affect RNAi efficiency. The methodology presented here can be adapted for studies of other fluorescent reporter genes, fluorescent knock-in genes, and by using end-point analysis strategies (for example, fixation and immunostaining) potentially any gene of interest.

A key challenge in efforts to understand and improve delivery methods for siRNA has been the unclear dose-response relationship of cytosolic siRNA, which we address here. Still unknown are the dose–responses for other nucleic acid therapeutics, including mRNA and CRISPR compounds, that can be delivered to the cytosol, where the latter has the added complexity of being a multi-component nucleic acid mixture, with distinct dose-response curves for each component. Extension of the method presented here to these and other nucleic acid compounds has the potential to define dose–response relationships and facilitate rational efforts to improve the efficiency of this emerging class of therapeutic molecules.

# Methods

## Cell culture and reagents

HeLa cells (CCL-2) were ordered from American Type Culture Collection and verified to be free from mycoplasma contamination. Cells were cultured in Dulbecco's Modified Eagle Medium (DMEM) (Hyclone, South Logan, UT, USA) supplemented with 10% fetal bovine serum (FBS, Gibco), 2 mM glutamine (Thermo Fisher Scientific, Waltham, MA, USA), 100 U mL$^{-1}$ penicillin, and 100 mg mL$^{-1}$ streptomycin and incubated at 37 °C, 5% $CO_2$. Prior to plating, cells were stained with trypan blue (Gibco) and counted with a Countess Automated Cell Counter (Invitrogen, Carlsbad, CA, USA) to obtain cell viability and concentration.

For all live-cell imaging experiments, cells were seeded $4-5 \times 10^4$ cells per well in 8-well Lab-Tek II chambered cover glass slides (Nunc, Rochester, NY, USA) and incubated overnight. Before image acquisition, cells were washed with PBS and incubated in an imaging medium (FluoroBrite DMEM (Gibco), 10% FBS, 2 mM glutamine, 2 mM HEPES) supplemented with $3.75 \times 10^{-3}$ μg mL$^{-1}$ Hoechst 33342 nuclear stain (Thermo Fischer Scientific) for 1–2 hours.

Cycloheximide (CHX) and dimethyl sulfoxide (DMSO) were both from Sigma. Custom-synthesized siRNA sequences were ordered from Integrated DNA Technologies. The following siRNA sequences were used: siGFP-1 sense: 5′-GGC UAC GUC CAG GAG CGC Atst-AF647-3′, siGFP-1 antisense: 5′-UGC GCU CCU GGA CGU AGC Ctst-3′, siGFP-2 sense: 5′-UGC UGC CCG ACA ACC ACU ACsC-AF647-3′, siGFP-2 antisense: 5′-UAG UGG UUG UCG GGC AGC AGsC-3′, siLuc sense: 5′-UCG AAG UAC UCA GCG UAA Gtst-3′, siLuc antisense: 5′-CUU ACG CUG AGU ACU UCG Atst-AF647-3′, siRNA-AF647(as) sense: 5′-GGC UAC GUC CAG GAG CGC Atst-3′, siRNA-AF647(as) antisense: 5′-UGC GCU CCU GGA CGU AGC Ctst-AF647-3′. Lowercase denotes deoxynucleotides and 's' indicates phosphorothioate linkage. Silencer Negative Control siRNA #1 (Invitrogen) was used as a negative control for Real-Time qPCR.

Primer pairs for PCR were from Sigma (eGFP) and Invitrogen (GAPDH). The following primers were used: eGFP forward: 5′-ACG TAA ACG GCC ACA AGT TC-3′, eGFP reverse: 5′-AAG TCG TGC TGC TTC ATG TG-3, GAPDH forward: 5′-CTG GGC TAC ACT GAG CAC C-3′, GAPDH reverse: 5′-AAG TGG TCG TTG AGG GCA ATG-3′.

HeLa cells stably expressing d1-eGFP were established by transfecting cells with a plasmid encoding d1-eGFP using Lipofectamine 2000 (Invitrogen) according to the manufacturer's protocol. For selection, transfected cells were grown and sub-cultured in a cell culture medium supplemented with 400 μg mL$^{-1}$ G418 (Sigma), followed by single-cell fluorescence-activated cell sorting using a BD FACSAria III cell sorter (Becton Dickinson, Franklin Lakes, NJ, USA) to obtain monoclonal cell lines. The d1-eGFP plasmid was from Invitrogen and constructed by cloning the d1-eGFP synthetic gene into a pcDNA3.3-TOPO vector backbone.

Transient expression of GFP-AGO2 in HeLa cells was achieved using a Neon Transfection System (Thermo Fischer Scientific). The 100-μL tip kit was used according to the cell-type specific protocol provided by the manufacturer. Plasmid encoding GFP-AGO2[36] was a gift from Phil Sharp (Addgene plasmid # 21981).

## Lipoplex formulation

Formation of siRNA lipoplexes was performed using a fixed volume of Lipofectamine 2000 (LF2000) and variable siRNA concentrations, with a final siRNA-lipoplex solution volume of 100 μL. siRNA and LF2000 were first diluted in OptiMEM before mixing and incubating for 20 min at room temperature. The following pmol:μL ratio of siRNA to LF2000 was used for siRNA lipoplex formulation for microscopy experiments: siGFP-1: 0.2:4, 0.5:4, 2:4, 10:4; siGFP-2: 0.2:4, 0.5:4, 2:4, 4:4, 10:4; siLuc: 2:4. For flow cytometry and RT-qPCR experiments, the following ratios were used: $2.6 \times 10^{-4}$:4, $1.3 \times 10^{-3}$:4, $6.4 \times 10^{-3}$:4, $3.2 \times 10^{-2}$:4, $1.6 \times 10^{-1}$:4, $8 \times 10^{-1}$:4, 4:4 and 20:4 pmol:μL siRNA to LF2000. For flow cytometry and RT-qPCR experiments, the added volume of siRNA-lipoplex solution corresponded to 10% of the final volume in the well. For microscopy experiments, 5, 10, 16.7, or 50 μL of the prepared siRNA-lipoplex solution was added to microscopy slide wells (final volume 500 μl), to vary the number of lipoplexes internalized by cells.

## Lipoplex size measurement

Lipoplexes of siGFP-1 were formulated with the following pmol:μL ratios of siRNA to LF2000: 0.2:4, 0.5:4, 2:4, and 10:4 (see lipoplex formulation section for more details). The lipoplex solution was diluted 1:10 or 1:25 in deionized water, added to a well in an 8-well Lab-Tek II chambered cover glass slide (Nunc), and allowed to settle on the glass slide for 40–50 min. Fluorescent and DIC single *z*-plane images of multiple lipoplexes were then acquired using an Airyscan confocal microscope. DIC images were acquired to confirm that the size of the lipoplexes was not affected by excessive fluorescence by the particle. Using raw fluorescent images, individual lipoplexes were identified and their pixel area was measured using a customized CellProfiler pipeline. A MATLAB script was further used to calculate the diameter of each structure by assuming the measured area was that of a circle.

## Live-cell imaging of siRNA release

HeLa cells stably expressing d1-eGFP or YFP-galectin-9 or transiently expressing GFP-AGO2 were plated in microscopy slides as described. Cells were transferred to a preheated microscopy incubation chamber, and 4–6 positions with sparse and evenly distributed cells were selected. Immediately before starting image acquisition, lipoplexes formulated with siGFP-1, siGFP-2, siLuc, or siRNA-AF647(as) were added dropwise to the medium. Typically, 5 or 10 μL of the siRNA-lipoplex solution was added to the well. For d1-eGFP knockdown experiments, one well was left untreated as a control and imaged using the same settings as treated cells. Two *z*-plane images with 4 μm *z*-spacing were acquired per position at 5 min intervals. The lower *z*-plane was set in the lower third of the cells (see confocal microscopy section for details). AF647-siRNA fluorescence was detected with an Airyscan detector while Hoechst 33342 and d1-eGFP fluorescence was detected with a PMT detector. Typically, images were acquired for 12–32 h for knockdown experiments and 6–8 h for galectin-9 recruitment experiments. AF647-siRNA bleaching was quantified in non-internalized glass-adhering lipoplexes.

For high-temporal resolution imaging of cytosolic siRNA release, a single *z*-plane set in the lower third of the cell was acquired at 5 s intervals (single position) and typically imaged for 25 min.

## Single-cell tracking and quantification

After time-lapse image acquisition, measurements of siRNA and eGFP fluorescence were performed in each cell. d1-eGFP and Hoechst 33342 images were denoised using the PureDenoise plugin[37] in Fiji, to improve segmentation. In each image frame, individual cells were then segmented, tracked, masked, and measured in CellProfiler using customized analysis pipelines, as described below. Fluorescence measurements were performed on raw images. Analyses were performed in the lower *z*-plane if not stated otherwise. In brief, segmented nuclei

were used for cell tracking and identified objects were labeled with unique identification numbers for downstream single-cell measurements. In experiments with siGFP, boundaries of cells become increasingly difficult to identify correctly, over time, using the eGFP signal, as the d1-eGFP expression decreases during a knockdown. Therefore, masks of segmented nuclei (insensitive to eGFP knockdown) were used for d1-eGFP fluorescence quantification. Comparisons of the d1-eGFP fluorescence intensity measured in nuclei alone and entire cells showed near identical results (Supplementary Fig. 18). Cytosolic AF647-siRNA fluorescence was measured with two types of segmentation masks, generated using cell boundaries identified in the d1-eGFP channel. A primary mask was used to detect sudden signal shifts, that is, for release event identification. A secondary mask, where the nuclear region was excluded, was used to quantify the cytosolic siRNA fluorescence intensity (as the quantification otherwise became highly sensitive to the exact position in z of the nucleus). In both cases, bright lipoplexes were identified in both imaging planes and excluded from the masks. Background signal intensity was calculated as the median pixel in the image after masking all identified objects (nuclei, cells, and lipoplexes). All measurements were median pixel values calculated in the corresponding segmentation masks. Values were background-corrected by subtraction of cell- and lipoplex-free background intensity values at each time point. Data were exported from CellProfiler as Excel files.

## Release event detection

A MATLAB script was used to automatically detect and validate sudden siRNA release events in single cells, using measurements from CellProfiler. In brief, the siRNA signal was evaluated in each frame to detect positive signal shifts. A shift in the signal intensity was considered as a potential siRNA release event if the shifted value and the five following measurements were all larger than the largest of (i) the sum of the mean intensity value of the previous three frames and three times the standard deviation of the previous ten frames, or (ii) the sum of the mean intensity value of the three previous frames and a fixed value. To reduce noise that might decrease the performance of the event-calling, a 5-frame moving median filter was applied to the siRNA signal before analysis. Adjustment of hyperparameters for the event detection algorithm was optimized empirically (visually) using a separate time-lapse imaging data set. After automated detection, all identified events were validated manually. Regions of interest (ROIs) were created containing individual cells with detected events, which were then concatenated into image panels for inspection in Fiji. Events were classified as true or false. A detected event was considered true if siRNA release could be visually observed at the frame of detection or 10 frames ahead, and false if no visual release was observed. For true events, visual release typically coincided with automated detection of the event or, could be perceived 1–3 frames after automated detection. Cells showing signs of siRNA release before the first identified event (for example, redistribution of cytosolic siRNA, see 60–120 min in Fig. 1c) were excluded from subsequent analysis. If multiple release events occurred in the same cell, detected events were evaluated for the second event.

## Sensitivity and specificity of cytosolic siRNA detection

HeLa cells stably expressing YFP-galectin-9 were imaged using live-cell microscopy for 8 h during treatment with lipoplex-formulated siGFP-1 at 2:4 pmol:μl ratio of siRNA to LF2000. Endosomal siRNA release was then automatically and manually detected in each cell. For manual event detection, each cell was observed frame by frame until a release event was visible or until the end of the acquisition. For event detection, both automated and automated combined with manual quality control, the procedure was performed as described above. De novo recruitment and colocalization of galectin-9 with siRNA-lipoplexes was then manually evaluated in all cells and the first galectin-9 positive

event was recorded. Cells located partially outside the image border at the time of siRNA release or galectin-9 recruitment were excluded. For the manual and automated detection of cytosolic siRNA release, cells were evaluated up until the first detected event. To determine the sensitivity and specificity of the cytosolic release detection to correctly identify the first siRNA release event in an evaluated cell, observations were classified as follows: cytosolic release events detected within five frames before or after galectin-9 recruitment to the releasing lipoplex were considered true positive observations. If no cytosolic siRNA was detected even though galectin-9 was recruited to a visible lipoplex, the observation was considered false negative. Observations were classified as true negative if no cytosolic siRNA release was detected and no recruitment of galectin-9 to lipoplexes could be observed, and false positive if the cytosolic release was detected in the absence of observable galectin-9 recruitment to the releasing lipoplex within five frames before or after the detection. Manual quality control was then performed of all siRNA release events identified by the automated detection algorithm, providing the opportunity to correct false positive events, that were then reclassified as true negative observations. This approach is analogous to the manual quality control of release events identified in experiments evaluating d1-eGFP knockdown (without galectin-9 reference). Release detection sensitivity was calculated as all true positive events divided by the sum of true positive and false negative observations. Specificity was calculated as true negative observations divided by the sum of true negative and false positive events.

### Absolute quantification of intracellular siRNA

Before live-cell imaging experiments where intracellular siRNA was quantified, a calibration sample with $1\,\mu M$ AF647-siRNA was prepared and imaged. Microscopy glass slides were prepared by adding $5\,\mu L\,1\,\mu M$ AF647-siRNA diluted in cytosol-mimicking buffer (CMB) ($15\,g\,L^{-1}$ BSA, $125\,mM$ KCl, $4\,mM$ $KH_2PO_4$, $14\,mM$ NaCl, $1\,mM$ $MgCl_2$, $20\,mM$ HEPES), or $5\,\mu L$ CMB only, to the center of slides. Droplets were covered with $18 \times 18\,mm\,1.5\#$ cover slips, approximately achieving a liquid column with a thickness similar to a cell, and imaged with the cover slip facing the objective. Three image stacks containing 30 $z$-planes with $0.5\,\mu m$ interval spacing were acquired through the center of the sample using identical microscopy settings (laser power, gain, pin-hole size, pixel size, etc.) as used during live-cell imaging. A MATLAB script was then used to calculate the mean pixel intensity of images in the $z$-stack to determine the full width at half maximum (FWHM) interval of all $z$-planes (that is, the interval of $z$-planes with a mean pixel intensity equal to or larger than the half maximum mean pixel intensity of a single $z$-plane image). A mean intensity projection across the $z$-dimension was created using all $z$-planes within the FWHM interval. A final mean reference image was then created from the $z$-projections of all three FWHM $z$-stacks. Pixel values were corrected for background fluorescence, and calculated using the image stacks of CMB samples in the same way as for the siRNA samples.

Cytosolic siRNA fluorescence, measured as the median pixel intensity in the cytosol and corrected for background fluorescence as described above, was corrected for uneven illumination (vignetting) and converted into absolute siRNA concentration by dividing measurements with the pixel value of the $1\,\mu M$ siRNA reference (calibration) image at the center of the evaluated cell. The maximal siRNA value (post-event) was used as a measure of the released siRNA quantity, except when model-based estimations were used (see below). The number of siRNA molecules per cell ($N_C$) was calculated from the measured cytosol concentration ($C$) and measured typical cytosol volumes ($V$) as $N_C = C \times V \times N_A$ where $N_A$ is the Avogadro constant.

To obtain a reference curve for the translation of fluorescence intensities within a typical cytosol volume to absolute concentrations, a standard curve was established using fourfold serial dilutions of AF647-siRNA between 1 and 1000 nM. The siRNA samples were diluted in CMB, prepared, imaged, and measured identically to the calibration sample described above.

To evaluate the linearity of the Airyscan detector at low fluorescence intensities, a standard curve was established using four-fold serial concentrations of AF647-siRNA between 0.016 and 250 nM. The siRNA samples were imaged in CMB and prepared in a single well of an 8-well Lab-Tek II chambered cover glass slides (Nunc), where the siRNA concentration was gradually increased while keeping the volume constant. Each concentration was imaged as single-plane images in five replicates, maintaining the $x$-, $y$-, and $z$-positions identical throughout the experiment. Fluorescence intensity was determined by calculating the mean pixel intensity of each replicate corrected for background fluorescence.

### Single-cell d1-eGFP expression analysis

After image acquisition, cells were tracked and quantified using Cell-Profiler as described above. A MATLAB script was then used for further analysis of single-cell measurements. In brief, cells with no or very low d1-eGFP expression at the beginning of the time-lapse acquisition were excluded from the analysis using a fixed value threshold. By evaluating changes in the size of cell nuclei, cells undergoing mitosis or apoptosis were identified. In cells where apoptosis was detected, measurements were masked starting 20 frames before cell death.

For d1-eGFP knockdown experiments, cells were excluded from analysis if they entered the field of view after the first three frames, to prevent including cells that may have had a prior undetected siRNA release event outside the frame. In addition, measurements after the detection of a second siRNA release event were masked, to limit all quantification of d1-eGFP knockdown to the effects of the first siRNA release event only. D1-eGFP measurements in cells with detected release events were corrected for bleaching and undetected release events, using measurements of cells in the same well that did not have any detected release event during lipoplex treatment (Supplementary Fig. 13). Single-cell measurements were shifted in time so that all detected release events were aligned at $t = 0$ and corrected for mitosis-induced eGFP fluctuations using AF647-siLuc control experiments (for details, see Supplementary Fig. 14 and Supplementary Note 3). Relative change in d1-eGFP expression was calculated by normalizing all values to the d1-eGFP intensity at the time of the siRNA release ($t = 0$). All cells were divided into equally sized quartiles or quintiles based on model-estimated magnitudes of siRNA release events.

For d1-eGFP half-life experiments with CHX, the d1-eGFP signal was corrected for bleaching using measurements of DMSO-treated control cells. The relative change in d1-eGFP expression was calculated by normalizing single-cell measurements to the d1-eGFP intensity in the first acquired frame ($t = 0$).

### Modeling of siRNA release and the knockdown kinetics

A full description and details of the mathematical models used to estimate siRNA release and d1-eGFP knockdown kinetics is available in Supplementary Note 1.

### d1-eGFP half-life

HeLa-d1-eGFP cells were plated in microscopy slides and incubated with an imaging medium supplemented with $3.75 \times 10^{-3}\,\mu g\,mL^{-1}$ Hoechst 33342. Cells were transferred to a preheated microscopy incubation chamber and 4−6 positions were selected in each of the two wells. Immediately before starting image acquisition, CHX (solubilized in DMSO) or DMSO only was added to either well with a final concentration of $50\,\mu g\,mL^{-1}$ and 0.05%, respectively. For each position, 2 $z$-plane images were acquired every 5 min for 5 h. The d1-eGFP fluorescence intensity was quantified and analyzed as described above.

## siRNA release in illuminated and non-illuminated cells

HeLa cells stably expressing YFP-galectin-9 were prepared in microscopy slides as described above and images of multiple positions were acquired using an Airyscan confocal microscope. Cells were then treated with lipoplex-formulated siGFP-1 at 2:4 pmol:µl ratio (siRNA:LF2000) and half of the positions were imaged every 5 min for 6 h (illuminated cells) during lipoplex treatment, while the other half remained non-illuminated. At the end of the acquisition, images of non-illuminated positions were immediately acquired again. The number of endosomal siRNA release events was then determined by counting the number of galectin-9 foci (colocalizing with siRNA-lipoplexes) in each cell (blinded with respect to illumination).

## Cytosol volume calculation

HeLa cells expressing YFP-galectin-9 (used as a cytosolic marker) were used to calculate the average cytosol volume. Cells were prepared in microscope-chambered glass slides, and nuclei were stained with Hoechst 33342 as described above. Images were acquired using a confocal microscope, obtaining z-stacks with 12 z-plane spaced 1 µm apart containing the entire cell volume. Cells were manually outlined in Fiji to create ROIs, typically in the bottom, middle and upper z-plane of individual cells, and then using the "interpolate ROIs" function to interpolate ROIs in the remaining planes. This procedure was repeated for cell nuclei. For each experiment, 20 representative cells were analyzed. Cell area measurements (per z-plane) were exported as Excel files. A MATLAB script was then used to calculate the volume of the cytosol by subtracting the volume of the nucleus from the total cell volume, given as femtoliters.

## Flow cytometry

HeLa-d1-eGFP cells were seeded $3 \times 10^4$ cells per well in a 48-well plate and incubated 24 h prior to siRNA treatment with either siGFP-1, siGFP-2, or no treatment (control). The growth medium was removed and HEPES-free imaging medium (FluoroBrite DMEM, 2 mM glutamine, 10% FBS) was added to each well. A five-fold serial dilution of siRNA was performed in OptiMEM. Diluted siRNA was mixed with an equal volume of LF2000 dissolved in OptiMEM (4 µl:50 µl) and incubated for 20 min at room temperature to form lipoplexes. Lipoplexes were then added to cells, 10% of the final volume, yielding final siRNA concentration between 0.26 pM and 20 nM with corresponding siRNA to LF2000 ratio between $2.6 \times 10^{-4}$:4 and 20:4 pmol:µL. Cells were incubated for 24 h, washed with PBS, detached by trypsinization, and resuspended in DMEM before transferring the medium to $12 \times 75$ polystyrene FACS tubes kept on ice. FACS tubes were centrifuged at $400 \times g$ for 5 min, the supernatant was discarded, and cells were resuspended in 0.5% BSA in PBS. The washing procedure was repeated, and cells were resuspended in 0.5% BSA in PBS and analyzed with an Accuri C6 Flow Cytometer (Becton Dickinson, Franklin Lakes, NJ, USA) using BD Accuri C6 Software v.1.0.264.21. Cells were gated inside scatter/forward scatter plots (Supplementary Fig. 19) and median fluorescence intensity was measured for each sample. HeLa wildtype cell fluorescence was subtracted to correct for background. For each experiment, samples were analyzed in three technical replicates. The mean of triplicates was calculated and normalized to untreated HeLa-d1-eGFP cells.

## Real-time qPCR

HeLa-d1-eGFP cells were seeded $1 \times 10^5$ cells per well in a 12-well plate and incubated for 24 h prior to siRNA treatment with siGFP-1 or negative control siRNA. Serial dilution of siRNA, lipoplex formulation and treatment were carried out as described above. Negative control siRNA was prepared at the same concentration as the highest siGFP dose. After 24 h lipoplex incubation, cells were washed with PBS, and RNA was extracted using GenElute Mammalian Total RNA Miniprep Kit according to the manufacturer's protocol. Complementary DNA was obtained using SuperScript III First-Strand Synthesis System (Sigma) with random hexamer primers running on a MasterCycler EpGradient 5341 thermal cycler (Eppendorf AG, Hamburg, Germany). Real-Time qPCR was performed on a StepOnePlus Real-Time PCR System using MicroAmp Fast 0.1 mL 96-well Reaction Plates (Applied Biosystems, Foster City, CA, USA) and SYBR Green JumpStart Taq Readymix (Sigma) for the qPCR reactions. GAPDH was used as a housekeeping gene, data were analyzed with StepOne Software v.2.3, and d1-eGFP expression fold-change relative to the control sample was calculated using the ΔΔCt method.

## Confocal microscopy

An inverted AxioObserver Z.1 LSM 710 epifluorescence confocal microscope with an Airyscan array detector unit (Carl Zeiss AG, Oberkochen, Germany) and equipped with a 40× Plan-Neofluar 1.3 numerical aperture (NA) oil-immersion objective was used for live-cell imaging acquisition. The FOV was set to 354,25 µm × 354,25 µm (full) for all experiments except for high temporal resolution imaging where the FOV was set to 177,12 µm × 177,12 µm. Scaling (per pixel) was kept constant. The pinhole was set to 2.15 Airy units (AU) in the software settings for the AF647-siRNA channel (the effective pinhole used by the Airyscan detector is 2.0 AU), and fully opened (599 AU) for the d1-eGFP channel. To arrange the two z-planes (Fig. 2a), the pinhole for the d1-eGFP channel was set to 1 AU to find the focal plane encompassing the largest area of the cells (z1) together with a focal plane 4 µm above (z2). The Airyscan Mode was Resolution vs. Sensitivity (R-S). A diode laser 405-30 (405 nm), a Lasos LGK 7812 argon laser (458 nm, 488 nm, 514 nm), and a HeNe633 laser (633 nm) were used as a light source. A stage-top incubator with an attached Temperature module, Heating Unit XL S and Heating Insert (Pecon), and $CO_2$ control system was used for all live-cell imaging experiments, operating at 37 °C and 5% $CO_2$. A Definite Focus module was used for auto-focus. The imaging system was operated under ZEN 2.1 (black edition).

## Software

CellProfiler[38] (cellprofiler.org) version 2.2.0 was used to set up customized pipelines for fluorescence quantification in single cells. MATLAB 2018b was used for post-processing and data analysis. GraphPad Prism 9 version 9.0.1 was used for graphs and to perform statistical testing. Fiji 2.1.0 and the plugin PureDenoise[37] (bigwww.epfl.ch/algorithms/denoise) was used for image processing, data analysis, and image visualization. Illustrator 2020 23.0.2 was used for the final figure design and composition.

## Statistics and reproducibility

Statistical tests and non-linear regressions were performed in GraphPad Prism 8 and 9. Non-linear regressions were fitted using "Sigmoidal, 4PL, X is log(concentration)" for sigmoidal curves, with Upper constraint = 1, Bottom constraint ≥0 with shared Hillslope when fitting multiple curves, "Log-log line−X and Y both log", with Y-intercept constant equal to 0 for log−log plotted standard curve, and "Straight line" for linear plotted standard curves. No statistical method was used to predetermine sample sizes. No data were excluded from the analyses and given the in vitro nature of the assays, experiments were not randomized.

## Reporting summary

Further information on research design is available in the Nature Portfolio Reporting Summary linked to this article.

## Data availability

Source data of all quantitative figures are provided in the Source data file. All data supporting the findings of this study are available at github.com/WittrupLab/CytosolQuant and at https://doi.org/10.6084/m9.figshare.c.5875859.v6. Source data are provided in this paper.

## Code availability

All custom computer code supporting the findings of this study is available at github.com/WittrupLab/CytosolQuant and the version of the code and data used in this publication can be accessed at https://doi.org/10.6084/m9.figshare.c.5875859.v6.

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

## Acknowledgements

This work was supported by grants to A.W. from the Swedish Society for Medical Research (SSMF), the Gunnar Nilsson Cancer Foundation, the Mrs. Berta Kamprad Foundations (FBKS-2020-33-308), the Winklers Foundation, the Governmental funding of clinical research within the National Health Services (A.L.F., YF2016), the Swedish Research Council (VR, 2020-02647), the Wallenberg Center for Molecular Medicine, Lund University, and the Knut and Alice Wallenberg foundation.

## Author contributions

A.W. conceived the study and supported funding. H.H., H.D.R., and A.W. designed the experiments, analyzed the data, and created figures. H.H. performed most of the experiments, and H.D.R., J.M.J., and H.C.E. performed some of the experiments. H.D.R. designed software tools for image processing. L.H. assisted in siRNA sequence selection. J.W. designed mathematical models and contributed to data analysis. W.Z. provided technical assistance. H.H., H.D.R., J.W., and A.W. wrote the paper with input from all authors.

## Funding

## Competing interests

The authors declare no competing interests.
