## [Peer Review File · Nature Communications]

Reviewers' Comments:

Reviewer #1:

Remarks to the Author:

The title of the paper is misleading. Absolute quantification suggests there is no room for uncertainty in the measurement of cytosolic delivery, I do not think this is the case. The image based analysis is really only semi-quantitative, so the title of the paper should be changed. Also reference to absolute quantification throughout the article should be revised.

The image based analysis is by its nature limited to the resolution of the chosen imaging technique. The resolution limit of this technique should be made clear in the manuscript. The lipoplexes used in this paper are quite large ($>1\mu\text{m}$) in comparison to other delivery systems like LNPs (which are 100nm) or even liposomes ($\sim 200\text{nm}$). The large size of the particles in this paper make it feasible to create a mask from the images to exclude the particles, however with particles $<250\text{nm}$ this would not be possible and would result in mis-identification of cytosolic delivery. The imaging technique described here would also not be able to resolve or differentiate between material trapped in a 100nm endosome from free cytosolic material and this limitation should be made clear in the manuscript.

In Figure 3c there are concentration measurements below the limit of detection that can be determined from the standard curve in Figure S1b. To report concentrations below 1nM, additional points on the standard curve would be required. Additionally, there are no error bars on Figure S1b. I would imagine there is quite a bit of error in the measurement if when the signal measured is only 1a.u. The uncertainty in the measurement at the low end of the standard curve would be high and should be determined and discussed.

I also found the discussion around the IC₅₀ of the siRNA's to be confusing. The quoted IC₅₀'s were determined using conventional knockdown experiments, but this shows that the knockdown is occurring at concentrations below the sensitivity of the assay.

There have been observations of light induced Endosomal disruption (PLoS One. 2011 Mar 14;6(3):e17732). How have the authors made sure that the release events that they have observed are not an artefact of the imaging?

Minor point - The references to the current state of art for detecting cytosolic delivery are not complete. A number of techniques to measure cytosolic delivery have not been mentioned and should be included (Angew. Chem. Int. Ed. 54, 15105–15108 (2015), J. Am. Chem. Soc. 140, 11360–11369 (2018), ACS Cent. Sci. 4, 1379–1393 (2018), Nature Communications, 11, 4482 (2020), ACS Chem. Biol. 2021, 16, 2, 293–309). Additionally reference 14 is a live cell assay, but is referred to as a non-live cell method.

Reviewer #2:

Remarks to the Author:

Hedlund et al. present data on the endosomal escape rate of dye-labeled siRNAs delivered with LNPs. The manuscript is well-written, exceptionally well-reasoned study and controlled. Figure 1h says it all, this confocal approach is extremely sensitive to, what appears to be, individual LNP releases of siRNAs into the cytoplasm. Previous groups have calculated that between 300 to 700 or more siRNAs are packaged per clinical grade 100 nm diameter ionizable LNP. The lipofection used here (non-clinical grade, fixed cationic charge), likely packages much more siRNAs per LNP, hence the large single release numbers observed. Overall, the studies are well controlled and the conclusions are supported by the primary data. I have several concerns listed below that, given the high caliber of this group, the authors should be able to readily address. In conclusion, this is an exciting study on detecting and counting the number of siRNAs escaping from an endosome via an LNP and the absolute amount of siRNAs required to reach a maximal RNAi response. It will, in all likelihood, be well recognized by the RNAi community.

Concerns:

1) To round out the introduction and put the current results in perspective in the discussion, the Brown et al, NAR, 2020, paper (ref #22 here) should be included in the introduction where 0.3% of GalNAc delivered siRNA escaped as should the He et al., NAR 2021, paper using nanoSIMS to evaluate ASO escape rates.

2) Fig. 3b. The authors need to include in their discussion and analysis the fact that LNP concentrations, such as [400 pM] used in Fig. 1a, are irrelevant because LNPs are not soluble particles, but are suspensions that precipitate, due to gravity, onto cells as a linear film and thereby dramatically increase the local amount on those cells. For Fig. 3b, the authors need to have a LNP lab measure the size of their particles. My guess is that the "0.2:4" ratio particle is the biggest and the "10:4" particle the smallest.

3) Fig 2. While reporting cytoplasmic concentration of siRNAs is valuable, I think the most important number is how many siRNAs actually were released, regardless of cell volume. Therefore, I suggest that the authors include both measurements.

Minor Concerns:

4) I think that the paragraph on p.6 starting out "As a model system to determine..." is meant to be under the subsequent heading (not above it).

Reviewer #3:

Remarks to the Author:

I have carefully read the entire manuscript and supplement.

Despite having gained good insight into the presented experiments and results, I will focus my review on the developed analysis pipeline only (the rest is too far away from my area of expertise to allow myself to be objective).

The image analysis pipeline consists of a sequence of heuristic steps that are loosely described, leaving quite some room for interpretation.

First, image denoising is performed with an existing Fiji plugin. The denoised images are then used to segment nuclei, cells, and bright puncta. These segmentations are subsequently used to create cytosolic masks (cells-nuclei-puncta), which are later used to quantify fluorescence expression levels.

The last sentence on page 13 remains unclear to me even after repeated reading.

Also, the imaging is performed on either 2 z-planes or in more (spaced at 1µm), but how the 2 planes are chosen does not seem entirely reproducible, is it?

In order to account for background fluorescence, the intensity value of the median-pixel (after masking) is subtracted. This seems to be an ok choice, but I wonder how sensitive this is to small changes in the segmentations that are used to create the masks. I wish the description would be slightly more technical (to enable full reproduction).

On the plus side, the authors made their analysis code available on GitHub. While it was not very easy for me to understand the code right away, I'm sure that others with a keen interest to reproducing or improving the presented results will eventually be able to do so.

Still, if I am not mistaken, large parts of the overall analysis pipeline is not contained in the GitHub repo but performed within Fiji, using multiple plugins, ROI annotations and interpolations, etc.

These aspects are hard to precisely reconfigure given the current manuscript.

The release event detection appears like another heuristic pipeline that does its job. Many choices (detection event criteria etc.) appear chosen to work and lack 'a good reason' to be as they are. If such reasons exist, they should be included, otherwise the authors might simply state that hyperparameters were chosen to lead to good results upon visual inspection (or some similar statement).

Additional to the automated event detection, a manual proofreading step is performed. Given the heuristic nature of the procedure, this seems to be a good idea.

The procedure described on page 15 states that the automated event detection is performed as described before. If that also contains the manual proofreading step, I wonder to what degree this can/should be called an 'automated' procedure.

In summary, I believe that the presented analysis pipeline is fit for measuring the presented data. I am not convinced that only given the current manuscript and the GitHub link it would be possible to fully reproduce the outlined procedure. Likely others could, though, come up with a similar setup that might lead to similar measurements. If this level of reproducibility is sufficient for this article is hard for me to judge. Certainly there exist many published works that describe their underlying analysis pipeline in similarly qualitative ways.

Rebuttal letter for “Absolute quantification and single-cell dose-response of cytosolic siRNA delivery”.

2022-05-23

Dear reviewers,

Thank you for taking the time to review our manuscript “Absolute quantification and single-cell dose-response of cytosolic siRNA delivery”. We appreciate the substantial effort that you have devoted into this submission and we kindly welcome the generous feedback that we have now tried to address in the revised version of the manuscript. For convenience, we will reply to each reviewer in the order that they were assigned and address each comment in the order that they were written.

To reviewer #1

- **The title of the paper is misleading. Absolute quantification suggests there is no room for uncertainty in the measurement of cytosolic delivery, I do not think this is the case. The image based analysis is really only semi-quantitative, so the title of the paper should be changed. Also reference to absolute quantification throughout the article should be revised.**

We understand your comments regarding the title, as “absolute” in general can have a connotation of being without error or perfect. However, given the current use of “absolute quantification” in the scientific literature and general scientific practice we believe that it is appropriate to refer to the method in the present manuscript as “absolute”. The usage of the term “absolute quantitative PCR” is non-controversial and can refer to both methods based on comparison to a standard curve (classical absolute qPCR and also our method) and to methods based on calibration-free techniques such as digital-PCR (see for example <https://www.thermofisher.com/se/en/home/life-science/pcr/real-time-pcr/real-time-pcr-learning-center/real-time-pcr-basics/absolute-vs-relative-quantification-real-time-pcr.html> for ThermoFisher Scientific’s descriptions). These “absolute quantification” methods are not without potential measurement errors but “absolute” instead refers to the fact that abundances can be expressed as a number of molecules and not just a relative intensity. Similar distinctions are made in the field of absolute mass-spectrometry¹ and microscopy². Obviously, the technical aspects of qPCR, MS and confocal microscopy differ quite substantially. However, the methods are at its core very similar, i.e. quantifying signal intensity and translating it into an absolute amount using a standard curve. We have clarified this reasoning in the manuscript. We believe this question to be a matter of terminology and are naturally open to alternative titles according to input from you as reviewers and editors but we still suggest that the current title should remain as it is. We have added a sentence to clarify the usage of “absolute” in the manuscript in the discussion (last paragraph on page 11).

- **The image based analysis is by its nature limited to the resolution of the chosen imaging technique. The resolution limit of this technique should be made clear in the manuscript. The lipoplexes used in this paper are quite large ($>1\mu\text{m}$) in comparison to other delivery systems like LNPs (which are 100nm) or even liposomes ($\sim 200\text{nm}$). The large size of the particles in this paper make it feasible to create a mask from the images to exclude the particles, however with particles $<250\text{nm}$ this would not be possible and would result in mis-identification of cytosolic delivery. The imaging technique described here would also not able to resolve or differentiate between material trapped in a 100nm endosome from free cytosolic material and this limitation should be made clear in the manuscript.**

This comment is certainly an important aspect and a key rationale for the feasibility of the technique that should be addressed in the manuscript. In principle the key factor determining if it is possible to mask the unreleased particles is the intensity differential versus the cytosol which in the case for lipoplexes is very large. In the case of eg LNPs this would be considerably more difficult. We have clarified this aspect in the discussion (middle of page 11).

- **In Figure 3c there are concentration measurements below the limit of detection that can be determined from the standard curve in Figure S1b. To report concentrations below 1nM, additional points on the standard curve would be required.**

This comment has been addressed and highlighted in the manuscript together with supplementary experiments. Our initial experimental setup included exchange of sample glass and volume for each sample in the reference curve. This introduces substantial background fluctuations that are not present in continuous imaging during cell imaging making variability exaggerated at low concentrations. Thus, to more accurately measure the linearity of the detector at very low signal levels we designed an experiment with a static sample chamber and fluid exchange with varying and low siRNA concentrations (0.016 nM – 250 nM). We show that the Airyscan detector is sensitive enough to detect AF647-siRNA fluorescent signal in a 0.016 nM siRNA dilution and maintain a linear agreement between fluorescence and concentration within the range of 0.016-250 nM of siRNA as demonstrated with lin-lin plots over several scales (suppl. Fig 1c). The low intensity release events measured during cytosolic siRNA delivery are more than ten-fold higher in concentration compared to the lowest dilution measured in the standard curve. Changes are made in the beginning of the second paragraph on page 4 (result section), in the two last paragraphs on page 18 (materials and methods section), and in Figure S1c. Thus, in conjunction with the standard curve in Figure S1b, which show linearity with a simple

setup for easy calibration, we believe that the sensitivity of the Airyscan detector to detect cytosolic siRNA delivery is sufficient and linear within the range of release events that have been quantified in this manuscript.

- **Additionally, there are no error bars on Figure S1b. I would imagine there is quite bit of error in the measurement if when the signal measured is only 1a.u.**

The error bars on Figure S1b are indeed missing and have now been provided. Thank you for pointing this out.

- **The uncertainty in the measurement at the low end of the standard curve would be high and should be determined and discussed.**

Data measurement uncertainty at the low end of the standard curve is now provided (see above) and discussed.

- **I also found the discussion around the IC50 of the siRNA's to be confusing. The quoted IC50's were determined using conventional knockdown experiments, but this shows that the knockdown is occurring at concentrations below the sensitivity of the assay.**

We have clarified the text (first paragraph on page 12). Conventional IC50 refers to the extracellular concentration added to cells while our method measures cytosolic concentrations that are related but also strongly determined by the siRNA:lipid ratio and the fact that the siRNA is in particulate matter and locally concentrated. The average extracellular concentration (of the particulate matter) can thus be outside of the detection range while it is still possible to detect the resulting intracellular signal, as we demonstrate.

- **There have been observations of light induced Endosomal disruption (PLoS One. 2011 Mar 14;6(3):e17732). How have the authors made sure that the release events that they have observed are not an artefact of the imaging?**

This comment has been addressed and highlighted in the manuscript together with supplementary experiments. Changes can be seen in the first paragraph on page 6 (result section), in the last paragraph on page 14 (materials and methods section) and in Figure S5. We show that lipoplexed AF647-siRNA inflicts similar number of endosomal damages on YFP-Galectin-9 HeLa cells irrespective of light exposure (similar in both continuously illuminated cells and cells that have not been illuminated and only end-point analyzed). Thus, the damages (and associated release events) are lipoplex-mediated endosomal disruption rather than an artefact of the imaging procedure.

- **Minor point - The references to the current state of art for detecting cytosolic delivery are not complete. A number of techniques to measure cytosolic delivery have not been mentioned and should be included (Angew. Chem. Int. Ed.54, 15105–15108 (2015), J. Am. Chem. Soc. 140, 11360–11369 (2018), ACS Cent. Sci. 4,**

1379–1393 (2018), *Nature Communications*, 11, 4482 (2020), *ACS Chem. Biol.* 2021, 16, 2, 293–309). Additionally reference 14 is a live cell assay, but is referred to as a non-live cell method.

Thank you for raising these points. The manuscript has been modified accordingly (first paragraph on page 3).

To reviewer #2

- **To round out the introduction and put the current results in perspective in the discussion, the Brown et al, NAR, 2020, paper (ref #22 here) should be included in the introduction where 0.3% of GalNAc delivered siRNA escaped as should the He et al., NAR 2021, paper using nanoSIMS to evaluate ASO escape rates.**

Good suggestions and important papers. Both are now cited in the introduction in relation to efforts to detect/quantify cytosolic nucleic acids.

- **Fig. 3b. The authors need to include in their discussion and analysis the fact that LNP concentrations, such as [400 pM] used in Fig. 1a, are irrelevant because LNPs are not soluble particles, but are suspensions that precipitate, due to gravity, onto cells as a linear film and thereby dramatically increase the local amount on those cells.**

This is an important point that was insufficiently discussed in the manuscript. We have added a section in the discussion on this (beginning of page 12).

- **For Fig. 3b, the authors need to have a LNP lab measure the size of their particles. My guess is that the “0.2:4” ratio particle is the biggest and the “10:4” particle the smallest.**

This comment has been addressed and highlighted in the manuscript together with supplementary experiments. Changes can be seen in the second paragraph on page 7 (results section), in the second paragraph on page 14 (materials and methods section), and in Figure S9. In brief, we investigated the particle size of “0.2:4”, “0.5:4”, “2:4”, and “10:4” siRNA:Lipofectamine (pmol, μ L) formulated lipoplexes using confocal microscopy imaging in conjunction with a custom-made CellProfiler pipeline to measure the size. We found that increased amount of siRNA relative to the lipid increased the overall particle size and widened the size distribution. Because of the large and very heterogeneous sizes of the particles they were difficult to measure correctly using classical Brownian motion tracking or DLS. However the size of the particles were well above the resolution limit of our Airyscan confocal system so this system could be used and was combined with label-free DIC imaging. The results do, furthermore, correspond well with Figure 3b where the release magnitude increases with an increased ratio of siRNA to lipid.

- **Fig 2. While reporting cytoplasmic concentration of siRNAs is valuable, I think the most important number is how many siRNAs actually were released, regardless of cell volume. Therefore, I suggest that the authors include both measurements.**

Intrinsically our method measure an apparent concentration but since the variability of cell size is relatively small this can be confidently translated into a number of molecules and we agree that this number is more intuitively understandable. Consequently, we have modified the manuscript to more clearly convey these values as well (last paragraph of page 8 to second paragraph of page 10).

- **I think that the paragraph on p.6 starting out “As a model system to determine...” is meant to be under the subsequent heading (not above it)**

Yes, this is a better order and the manuscript has been modified.

To reviewer #3

- **The image analysis pipeline consists of a sequence of heuristic steps that are loosely described, leaving quite some room for interpretation. First, image denoising is performed with an existing Fiji plugin. The denoised images are then used to segment nuclei, cells, and bright puncta. These segmentations are subsequently used to create cytosolic masks (cells-nuclei-puncta), which are later used to quantify fluorescence expression levels. The last sentence on page 13 remains unclear to me even after repeated reading.**

We understand that the sentence you mentioned had a peculiar wording. Thank you for pointing that out. The paragraph has now been re-worded and is hopefully more clear. See last paragraph on page 15.

- **Also, the imaging is performed on either 2 z-planes or in more (spaced at 1um), but how the 2 planes are chosen does not seem entirely reproducible, is it?**

This aspect has been clarified in the manuscript. Changes can be seen in the third paragraph on page 20. Here, we describe the procedure of setting up the 2 z-planes in more detail. The plane selection is quite reproducible but ultimately not necessary to be completely identical each time as the signal in the cell is highly homogenous initially (and the analysis plane can thus differ somewhat). However, we want to image as large a section of the cells as possible to optimize sensitivity and noise level and this is what we strive for.

- **In order to account for background fluorescence, the intensity value of the median-pixel (after masking) is subtracted. This seems to be an ok choice, but I wonder how sensitive this is to small changes in the segmentations that are used to create the masks. I wish the description would be slightly more technical (to enable full reproduction).**

This comment has been addressed. We have provided the full CellProfiler pipeline, an example data set and a manual for the complete analysis pipeline, including steps in the CellProfiler pipeline. This should be sufficient to enable full reproduction. Regarding the sensitivity to variations in segmenting masks, given the time-lapse characteristic of the data, all cells are segmented multiple times with slightly varying masks. During method development it was noted, as the reviewer alludes to, that changes in segmentation had substantial effects on *mean* fluorescence measurements giving rise to substantial noise and variability. By instead analyzing the *median* pixel intensity most of this variability was muted and noise lowered. Thus, segmentation induced errors were minimized as for example seen in the very clean traces for a typical high intensity release example in fig 2d. Instead, measurement errors for low intensity release events are substantially larger than any segmentation induced errors (which would show up also in high release events).

- **On the plus side, the authors made their analysis code available on GitHub. While it was not very easy for me to understand the code right away, I'm sure that others with a keen interest to reproducing or improving the presented results will eventually be able to do so. Still, if I am not mistaken, large parts of the overall analysis pipeline is not contained in the GitHub repo but performed within Fiji, using multiple plugins, ROI annotations and interpolations, etc. These aspects are hard to precisely reconfigure given the current manuscript.**

This comment has been addressed and the analysis pipeline is now available in the GitHub repo. As briefly mentioned above, it includes the CellProfiler pipeline, the custom-made app (CytosolQuant), an example data set and an attached manual to provide step-by-step guidance.

- **The release event detection appears like another heuristic pipeline that does its job. Many choices (detection event criteria etc.) appear chosen to work and lack 'a good reason' to be as they are. If such reasons exist, they should be included, otherwise the authors might simply state that hyperparameters were chosen to lead to good results upon visual inspection (or some similar statement).**

The hyperparameters were determined empirically to "work" as determined visually. This has been clarified (first paragraph of page 16).

- **The procedure described on page 15 states that the automated event detection is performed as described before. If that also contains the manual proofreading step, I wonder to what degree this can/should be called an 'automated' procedure.**

This is done both with and without the manual quality control. The text has been clarified (last paragraph of page 16).

References

1. Gerber, S. A., Rush, J., Stemman, O., Kirschner, M. W. & Gygi, S. P. Absolute quantification of proteins and phosphoproteins from cell lysates by tandem MS. *Proc. Natl. Acad. Sci.* **100**, 6940–6945 (2003).

2. Chatzimichail, S., Supramaniam, P. & Salehi-Reyhani, A. Absolute Quantification of Protein Copy Number in Single Cells With Immunofluorescence Microscopy Calibrated Using Single-Molecule Microarrays. *Anal. Chem.* **93**, 6656–6664 (2021).

Reviewers' Comments:

Reviewer #1:

Remarks to the Author:

The authors have made some changes to the manuscript, but unfortunately I still don't think the paper is suitable for publication.

My principal concern with this manuscript is that small endosomal compartments cannot be distinguished from the cytosol. This is a fundamental limitation of image based techniques to measure endosomal escape. The authors say in the discussion that they "assume the siRNA molecules are primarily free molecules within the cytosol". Even if this assumption were to be correct for lipofectamine (which I think is a separate issue), the assumption does not hold for most delivery systems. This limits the use of this technique to studying large particles (>1 μm) that do not fragment and deliver 100% of their cargo to the cytosol. This narrows the applicability of the technique and as such I do not think it is appropriate for a general science audience of Nature Communications.

There is a large amount of hyperbole used in the manuscript which I don't think are backed up by the data. I still disagree that the technique is quantitative. qPCR does not make the same large number of assumptions that are required for this technique. If the assumption that 100% of the siRNA are free molecules in the cytosol is incorrect, then the numbers are not quantitative. Similarly, claiming the method is robust when the technique will only work for a narrow range of conditions (particles larger than 1 μm , 100% release into the cytosol, no fragmentation of particles when internalised etc) I think is a stretch.

I disagree with the authors interpretation of Supplementary Movie 1 and Figure 1a. The movie is of high quality and clearly shows that siRNA has moved into vesicles. I don't understand why after being released into the cytosol, the free siRNA would move into vesicles.

I'm not sure I 100% agree with the limitations given for the existing sensor methods for continuous sensor strategies. The limits of detection for splitGFP and other fluorescent techniques would be similar to the fluorescence technique outlined here. The sensor techniques are not limited by the resolution of imaging. Also, a limitation given for the FCS technique is that it "relies on subjective selection of cytosolic regions". I assume this means to avoid small vesicles that contain siRNA. However this imaging technique assumes that all signal that isn't from the original particle is free in the cytosol. In my opinion this is a bigger limitation that the selection of cytosolic regions required for FCS.

The manuscript lacks robust statistical analysis. By this I mean, the uncertainty in the measured numbers is not quoted. I am not an expert in modelling, but from Sup Fig 6, none of the models appear to fit the data well.

A minor comment, the discussion at the start of the paper about lipid nanoparticles suggests that this technique could be used to measure endosomal escape from these type of particles. This limitation isn't mentioned until the end of the discussion.

Reviewer #2:

Remarks to the Author:

The authors have satisfactorily addressed all of my concerns, which were minor to start with. This is an excellent study from the preeminent group in this area and one that will in all likelihood be highly cited as a Nature Communications publication.

Reviewer #3:

Remarks to the Author:

All concerns I have voiced earlier have now been addressed.

Rebuttal letter for “Single-cell quantification and dose-response of cytosolic siRNA delivery”.

2022-11-10

Dear reviewers,

Once again, thank you for taking the time to review the revised version of our manuscript, now with the title “Single-cell quantification and dose-response of cytosolic siRNA delivery”. We appreciate the valuable input from all reviewers, which has substantially improved the quality of the manuscript. We believe we have addressed all the concerns raised by reviewer #1 in the last round of review. In particular, we believe we have substantially clarified the manuscript and the formal support for the quantitative aspects. We also thank reviewer #1 for alerting us to the fact that some of our reasoning behind our firm conviction that we are measuring cytosolic siRNA was not sufficiently clear in the previous versions of the manuscript. We believe the reasoning now is easier to follow and strongly reinforced by new data in the manuscript.

To reviewer #1

- **The authors have made some changes to the manuscript, but unfortunately I still don't think the paper is suitable for publication. My principal concern with this manuscript is that small endosomal compartments cannot be distinguished from the cytosol. This is a fundamental limitation of image based techniques to measure endosomal escape.**

We have clarified a number of points in the following responses to address the concerns raised regarding the distinction between cytosolic and endosomal (or otherwise localized) fluorescence for siRNA quantification. We agree this is a key aspect of the present study, and it has in fact served as motivation for the whole project and method development.

When we measure the cytosolic siRNA, we perform several steps to obtain an as “pure” cytosolic fluorescence readout as possible. First, bright structures (lipoplexes) are segmented and excluded from subsequent measurements, including the surrounding haze. This removal is not dependent on particle size but rather the **intensity** of lipoplexes, that is typically substantial. However, a small (even below the resolution of the imaging system) but bright vesicle would still be detected in a few pixels, enabling removal.

Second, we measure the **median** pixel intensity in the cytoplasmic region of the cell. This is important as it leads to strong suppression of any remaining endosomal (focal) signal. We have emphasized this in the results section.

Third, in our quantitative efforts we analyze and detect a sudden **increase** in the siRNA signal. This means that any remaining lipoplex-signal (from *e.g.*, small but

numerous vesicles) or uneven background will generally be present in multiple frames, and is corrected for during background/zero-offset correction.

In the end the question is to what extent we by these means are successful in isolating a representative cytosolic signal, reflecting endosomal escape, and to what extent the signal change is due to artefacts, noise and non-cytosolic signal. The central message of the manuscript is that we are indeed successful in probing the cytosol with high sensitivity and specificity, and we have now tried to further clarify this. The new **Figure 3b** demonstrates a clear and sudden increase in cytosolic siRNA signal in virtually all 187 analyzed cells after appearance of a galectin-9 positive membrane damage. Very low variability of the cytosolic siRNA signal before galectin-9 damage (i.e. background) is also evident, with only a handful of traces exhibiting more substantial signal. Thus, the measured cytosolic signal is very specific for endosomal escape associated with membrane damage. The new **Supplementary Movie 3** illustrates the typical strict correlation between membrane damage (galectin-9 recruitment), pan-cytoplasmic siRNA distribution and the highly specific cytosol-filtered siRNA signal.

Finally, we derive time-resolved dose-dependency in knockdown response as well as a clear difference in estimated intracellular IC50 for two different siRNA sequences from the cytosolic siRNA measurements. How this data could be obtained if we do not measure cytosolic siRNA after endosomal escape is difficult to explain.

- **The authors say in the discussion that they “assume the siRNA molecules are primarily free molecules within the cytosol”. Even if this assumption were to be correct for lipofectamine (which I think is a separate issue), the assumption does not hold for most delivery systems.**

We have changed the wording to better reflect the message we want to convey, i.e. the siRNA is not released as large supramolecular complexes which can exhibit non-linear fluorescence intensities (quenching). The new **Figure 1b** and **Supplementary Movie 2** show high-temporal resolution imaging of the endosomal release event with a clear diffusion front moving throughout the cell over the course of approximately 20 s, which is consistent with a 14 kDa siRNA but not consistent with large supramolecular aggregates or intact vesicles. At early time-points after release the siRNA signal is always homogenous with no hints of aggregates or large complexes (as shown in the new **Supplementary Figure 1** and also in **Supplementary Movie 1**). Release of siRNA from lipid nanoparticles is in many aspects (mechanism, compartment, galectin-response, inefficiency) quite similar to lipoplexes (but much smaller). How other delivery systems behave at the time of release is less clear and beyond the scope of this study, but we believe the presented method is a powerful strategy to study this question.

- **This limits the use of this technique to studying large particles (>1 μm) that do not fragment and deliver 100% of their cargo to the cytosol. This narrows the applicability of the technique and as such I do not think it is appropriate for a general science audience of Nature Communications.**

There is no strict size limit and an extension to smaller particles are conceivable with improved fluorescence microscopy techniques, which has been further clarified in the text. There is no evidence suggesting particle fragmentation to be an issue. Most likely, the fragmented particles would still be masked and if not, fragmented particles would have an insignificant effect on the median pixel intensity of the whole cell. We do not require 100% delivery into the cytosol – quite the contrary. See for example the new **Supplementary Movies 2 and 3** where all the releasing particles remain relatively intact because of only a limited release fraction.

Importantly, we also emphasize that a key benefit of the paper is the ability to obtain an intracellular dose-response curve for siRNA which can be used as a basis for further studies with other delivery strategies. This is of substantial general interest in the field of oligonucleotide therapeutics.

- **There is a large amount of hyperbole used in the manuscript which I don't think are backed up by the data. I still disagree that the technique is quantitative. qPCR does not make the same large number of assumptions that are required for this technique. If the assumption that 100% of the siRNA are free molecules in the cytosol is incorrect, then the numbers are not quantitative. Similarly, claiming the method is robust when the technique will only work for a narrow range of conditions (particles larger than 1µm, 100% release into the cytosol, no fragmentation of particles when internalised etc) I think is a stretch.**

The quantitative validity of the method is indeed important as raised by the reviewer. The technical aspects of qPCR are undoubtedly very different from the technique that we are using, and thus assumptions will differ. As we state in the manuscript, the comparison is referring to the utility of a reference curve to elucidate absolute quantitative measurements. The use of fluorescence intensity values from a calibrated point-scanning confocal microscope to calculate absolute concentrations has recently been shown and this reference has been added to the manuscript (new ref. 26). Regarding the generalizability, as we describe above, we do not assume 100% free molecules (just not large supramolecular aggregates, which our data refute), particle size is not a strict limit for the method, 100% release is not required and potential fragmentation would not degrade or otherwise corrupt the measured cytosolic intensity. Thus, these aspects do not impair the generalizability of the method.

However, we also claim single-cell quantitative validity based on the high model fit at release magnitudes above ~0.5–1 nM cytosolic siRNA. We realize the support for this claim might have been somewhat opaque in the previous version of the manuscript. For this reason, we have now included a formal analysis of uncertainty (new **Supplementary Note 2** and **Supplementary Figure 12**) for the quantification reliability of the individual release quantifications. In this analysis we have included potential uncertainty from errors in reference curve measurements, fluorescence bleaching variability and background variability, in addition to the error from the uncertainty of the model fitting. Importantly, this analysis demonstrates that the

uncertainty (estimated standard deviation or 67% confidence interval) is well below 20% for the vast majority of quantifications, making single-cell analysis feasible and appropriate. While the mathematics of the uncertainty analysis is not overly complicated, the essence can be summarized as: Given the low variability of the reference curve measurements (high linearity and reproducibility), the quantification of a siRNA release event exhibiting a high fit to a stereotypical release model (with a sudden fluorescence increase followed by a slow exponential decay, e.g. **Figure 3e**), has a high measurement confidence, i.e. the true cytosolic release amount is close to the quantified value. Uncertainty increases at lower release magnitudes, driven by lower model fit due to more dominant noise in the measurements.

We have furthermore changed the title to not give the impression that our measurements are absolute in the sense of being without error. We have also removed the term *absolute* throughout the manuscript where we have deemed it superfluous and not necessary for the meaning. It is still used where necessary to distinguish from non-absolute (i.e. relative) measurements.

- **I disagree with the authors interpretation of Supplementary Movie 1 and Figure 1a. The movie is of high quality and clearly shows that siRNA has moved into vesicles. I don't understand why after being released into the cytosol, the free siRNA would move into vesicles.**

We have added several new pieces of evidence that the phenomenon we are observing with sudden pan-cytoplasmic distribution of siRNA is indeed cytosolic siRNA after endosomal release. The new **Figure 1b** and **Supplementary Movie 2** shows high-temporal resolution imaging of the endosomal release event with a clear diffusion front moving through the cell over the course of approximately 20 s. The slower appearance (over several minutes) of cytoplasmic foci after an initial phase of non-granular siRNA signal is also evident here as well as in multiple examples in **Supplementary Figure 1**. Moreover, in new experiments we demonstrate, by attaching a fluorophore to the active strand of the siRNA, that a subset of the active strands redistributes to pre-existing cytosolic AGO2-positive structures after release (new **Figure 1d**). That the siRNA (both the active and inactive strands) redistributes into cytoplasmic foci after an initial phase of homogenous distribution is thus expected and further evidence of cytosolic localization.

To summarize, the conclusion that we are indeed detecting and measuring cytosolic siRNA thus rests on the following evidence:

1. Galectin-9 is recruited to vesicles containing lipoplex siRNA (indicating membrane damage to the vesicle) at the very time cytoplasmic siRNA signal is detectable (**Figure 1b, 3b, Movie S2, and Movie S3**).
2. Diffusion of siRNA throughout the cytoplasm (nongranular) directly after galectin-9 recruitment, **Figure 1b and Movie S2**.

3. Subsequent siRNA redistribution over the course of minutes into cytoplasmic foci, such as to AGO2-positive structures (**Figure 1d**, **Suppl. Figure 1**).

4. Dose-dependent knockdown of a target protein (**Figure 5**).

- **I'm not sure I 100% agree with the limitations given for the existing sensor methods for continuous sensor strategies. The limits of detection for splitGFP and other fluorescent techniques would be similar to the fluorescence technique outlined here. The sensor techniques are not limited by the resolution of imaging. Also, a limitation given for the FCS technique is that it "relies on subjective selection of cytosolic regions". I assume this means to avoid small vesicles that contain siRNA. However this imaging technique assumes that all signal that isn't from the original particle is free in the cytosol. In my opinion this is a bigger limitation that the selection of cytosolic regions required for FCS.**

One key advantage with our method compared to for example splitGFP or other sensor-based methods is the high temporal resolution that we achieve here. This is highly valuable given the rapid dynamics of the biological processes during endosomal escape and cytosolic redistribution or sorting. SplitGFP fluorescence maturation is on the order of minutes to hours. Chemical fluorophore-based methods are also much easier to modify, using alternative wavelengths (spectra) for various multiplexed read-outs, and there is a wide variety of commercial providers. Additionally, the fluorophore is small enough to attach on the functional strand of the siRNA, making it possible to track the pharmacologically active molecule (new **Figure 1d**). The spectral flexibility (e.g. far-red fluorescent siRNA) combined with the rapid detection kinetics is precisely what is used for our analysis of knockdown of the well-established destabilized d1-eGFP reporter gene. The corresponding experiment would be difficult to perform using splitGFP, where the influence of sensor-expression-level would also be a quantitatively complicating factor.

For FCS the challenges of selecting the right area to image is basically equivalent to selecting the area to measure the fluorescence intensity using our procedure. The major difference is the ease by which, here, this selection can be automated to generate data from large numbers of cells. For FCS based methods it would be extremely challenging to get continuous FCS traces from the hundreds of cells per experiment we generate. This is crucial if one wants to correlate the measured release amounts to other biological properties that might exhibit substantial biological variability, as for example knockdown response. In these cases, a large number of cells need to be analyzed. This is the reason we are able to obtain the detailed time-resolved dose-dependency curves of knockdown responses that have not been measured previously.

- **The manuscript lacks robust statistical analysis. By this I mean, the uncertainty in the measured numbers is not quoted. I am not an expert in modelling, but from Sup Fig 6, none of the models appear to fit the data well.**

We acknowledge that uncertainty measures were in some cases omitted in the previous manuscript for improved readability, especially where we also presented ranges derived from cells of different size. We have now changed the data presentation to consistently include uncertainty ranges where applicable and data distribution ranges where more appropriate in the results section. Exemplary mentions of data in the discussion section (in an order of magnitude fashion) is for readability presented with mid-point estimates and a tilde (~) symbol. We fully agree that these changes have clarified the manuscript.

Supplementary Fig. 6 (now **Supplementary. Fig. 7**) is an example of a noisy trace to demonstrate that the simpler model with a single release event is selected despite a lower R^2 (0.71) compared to the more complex double release event model ($R^2 = 0.80$). The single release model is selected on the basis of the BIC (Bayesian Information Content) to better reflect the data (avoiding over-fitting of the more complex model).

We have also included a formal uncertainty analysis to give more comprehensive statistical estimates of the uncertainty of the individual siRNA quantifications (new **Supplementary Note 2** and **Supplementary Figure 12**).

A minor comment, the discussion at the start of the paper about lipid nanoparticles suggests that this technique could be used to measure endosomal escape from these type of particles. This limitation isn't mentioned until the end of the discussion.

We have added a clarification to the introduction that a key outstanding question hampering studies of endosomal escape using various delivery strategies is the lack of a well-defined dose-response relationship of intracellular siRNAs – precisely what we achieved to establish in the present study. The fact that we are successful in directly imaging cytosolic siRNA is facilitated by the large and bright lipoplexes, but as we have also clarified in the discussion, it is conceivable that the current rapid development of fluorescence microscopy techniques (higher resolution, sensitivity and speed) will enable direct imaging of LNP-mediated RNA delivery as well. Here, the key findings are that it is possible to do direct imaging of pharmacologically relevant amounts of cytosolic siRNA **and** that this imaging can be used to establish a time-resolved dose-response relationship.

Reviewers' Comments:

Reviewer #1:

Remarks to the Author:

The authors have made a number of changes to the paper and included some interesting new data, but unfortunately I remain unconvinced by the fundamental premise of their image based approach to measure endosomal escape. In my opinion, the masking approach to remove signal from the particles in cells will fundamentally limit this approach to only analysing particles that are substantially larger than the resolving power of the microscope, and will not be able to distinguish between cytosolic signal and cargo that has been released from the large particles into smaller endosomal compartments.

I understand that the authors are masking based on intensity rather than pixel area, but this does not overcome the limitation of the resolving power of the microscope. For example, a single 100nm vesicle will conservatively occupy only 3% of the voxel volume. I do not think conventional confocal imaging combined with their masking will be able to eliminate these single pixels, and the authors have not shown that their system is capable of doing this. In fact their masking does not remove the localised foci in the cytosol that occur after the burst release of signal from the endosomes.

The masks shown throughout this manuscript only show very large particles being masked out. Additionally, the inherent noise from sampling which is present in the images means that identifying the point at which the intensity fluctuation in the image is the result of a sub-resolution vesicle vs normal poisson noise from free cargo is very challenging. This will require an arbitrary threshold to be set.

The sudden increase in cytosolic signal corresponding to gal-9 recruitment is interesting, compelling and well performed. This is certainly an interesting phenomena that there is subsequent recruitment to cytosolic foci. The authors have demonstrated well that any subsequent signal measured is likely to be from this endosomal escape event. However, this doesn't overcome the limitations of the imaging approach in the paper (i.e. that everything small in the cytosol is counted as endosomal escape).

My principal criticism and concern with this paper is that the image based approach will only work for a limited number of circumstances. In my opinion they are:

1. The particle must be greater in size than the resolving power of the microscope. Being conservative, this limits the approach to particles >500nm (but practically larger than that). Other than lipofectamine, most commonly used delivery systems (and certainly those used in vivo) are 200nm or smaller, so I do not believe this technique will be useful for them. If it can be used for these systems, then the authors need to demonstrate it.
2. Even if the particle is larger than 500nm, the endosomal escape from the particle needs to be a burst event, and not release of cargo in the endosome where there could be transport/fragmentation of the larger endosomal compartment into smaller ones.

For the paper to be published, these limitations should be made very clear. It also narrows the scope and applicability of the technique, which means I do not think it is of interest for a wide audience of Nature Communications. I would be concerned that if published in the current form, other researchers would use this technique to measure the endosomal escape from LNPs and liposomes (both of which are below the resolving power of a confocal microscope) and leading to incorrect determination of endosomal escape.

Reviewer #4:

Remarks to the Author:

In the revised manuscript by Hedlund et al., the authors have gone to great extents to address and satisfy all of the reviewer's comments. The authors are attempting to tackle the very difficult problem of quantifying endosomal escape with a highly sophisticated approach and analytics.

Regarding Reviewer #1's point that the endosomal escape from the particle needs to be a burst event, and not release of cargo in the endosome where there could be transport/fragmentation, prior microscopic work from Prof. Zerial's lab on Patisiran (NBT, 2013) argues that LNP fragmentation is not occurring within endosomes and that LNPs release in a burst, just as Hedlund et al. detect here.

Regarding Reviewer #1's point on the limitation of light microscopy to ascertain what has escaped the endosome and where it has gone, s/he is correct on that point. The light limitation requiring >500 nm particles excludes LNPs (<150 nm diameter, generally ~80-100 nm), which includes the mRNA vaccines and Alnylam's Patisiran siRNA, and any academic LNP or liposome work. However, I strongly disagree with Reviewer #1 that this kills the study for publication because determining the amount of escape from the endosome is an extremely difficult and crucial problem to address. However, this is a significant methodological deficiency that the authors need to ensure is thoroughly discussed in the text so that the non-specialist reader will appreciate the limitation of their study.

And I also disagree with Reviewer #1 that this is not appropriate for a general audience. Quite the opposite, I think this study brings to light the complexities, deficiencies and unknowns of LNP delivery, something that most have likely physically experienced via the COVID mRNA vaccines.

Regarding Reviewer #1's concerns of hyperbole, I agree with the concern and the authors have satisfactorily toned down and tightened the text.

Overall, Hedlund et al. are quantitatively tackling a very difficult problem. They have bent over backwards to address all of the reviewers primary and secondary review concerns. Reviewer #1's concerns, though still quite negative, have resulted in a greatly improved manuscript. The study is important for both the specialist and general audience, and as such, the manuscript should be accepted.

Response to Reviewer's comments

We thank both reviewers for the, once again, very solid reading of the manuscript and the constructive critique. Our responses are in bold.

Reviewer #1 (Remarks to the Author):

The authors have made a number of changes to the paper and included some interesting new data, but unfortunately I remain unconvinced by the fundamental premise of their image based approach to measure endosomal escape. In my opinion, the masking approach to remove signal from the particles in cells will fundamentally limit this approach to only analysing particles that are substantially larger than the resolving power of the microscope, and will not be able to distinguish between cytosolic signal and cargo that has been released from the large particles into smaller endosomal compartments.

I understand that the authors are masking based on intensity rather than pixel area, but this does not overcome the limitation of the resolving power of the microscope. For example, a single 100nm vesicle will conservatively occupy only 3% of the voxel volume. I do not think conventional confocal imaging combined with their masking will be able to eliminate these single pixels, and the authors have not shown that their system is capable of doing this. In fact their masking does not remove the localised foci in the cytosol that occur after the burst release of signal from the endosomes.

The masks shown throughout this manuscript only show very large particles being masked out. Additionally, the inherent noise from sampling which is present in the images means that identifying the point at which the intensity fluctuation in the image is the result of a sub-resolution vesicle vs normal poisson noise from free cargo is very challenging. This will require an arbitrary threshold to be set.

A strong signal emanating from a small source (even a single molecule) will be diffracted and spread over a larger area corresponding to the resolving power of the microscope. Depending on the intensity and cut-off for what intensity is considered positive (the tail of the airy-disc) this can be substantially larger than the "resolution" of the system and will typically be several pixels (at Nyquist sampling). In our case, the particles are quite large and very intense and thus easy to mask. We have reinforced this limitation of the technique in the discussion section.

The sudden increase in cytosolic signal corresponding to gal-9 recruitment is interesting, compelling and well performed. This is certainly an interesting phenomena that there is subsequent recruitment to cytosolic foci. The authors have demonstrated well that any subsequent signal measured is likely to be from this endosomal escape event. However, this doesn't overcome the limitations of the imaging approach in the paper (i.e. that everything small in the cytosol is counted as endosomal escape).

My principal criticism and concern with this paper is that the image based approach will only work for a limited number of circumstances. In my opinion they are:

1. The particle must be greater in size than the resolving power of the microscope. Being conservative, this limits the approach to particles >500nm (but practically larger than that). Other than lipofectamine, most commonly used delivery systems (and certainly those used in vivo) are 200nm or smaller, so I do not believe this technique will be useful for them. If it can be used for these systems, then the authors need to demonstrate it.
2. Even if the particle is larger than 500nm, the endosomal escape from the particle needs to be a burst event, and not release of cargo in the endosome where there could be transport/fragmentation of the larger endosomal compartment into smaller ones.

There is not a firm size limit but rather a limit that the intensity has to be sufficient for intensity-based masking (i.e. clearly above poisson noise). We have clarified this in the discussion.

For the paper to be published, these limitations should be made very clear. It also narrows the scope and applicability of the technique, which means I do not think it is of interest for a wide audience of Nature Communications. I would be concerned that if published in the current form, other researchers would use this technique to measure the endosomal escape from LNPs and liposomes (both of which are below the resolving power of a confocal microscope) and leading to incorrect determination of endosomal escape.

Reviewer #4 (Remarks to the Author):

In the revised manuscript by Hedlund et al., the authors have gone to great extents to address and satisfy all of the reviewer's comments. The authors are attempting to tackle the very difficult problem of quantifying endosomal escape with a highly sophisticated approach and analytics.

Regarding Reviewer #1's point that the endosomal escape from the particle needs to be a burst event, and not release of cargo in the endosome where there could be transport/fragmentation, prior microscopic work from Prof. Zerial's lab on Patisiran (NBT, 2013) argues that LNP fragmentation is not occurring within endosomes and that LNPs release in a burst, just as Hedlund et al. detect here.

Regarding Reviewer #1's point on the limitation of light microscopy to ascertain what has escaped the endosome and where it has gone, s/he is correct on that point. The light limitation requiring >500 nm particles excludes LNPs (<150 nm diameter, generally ~80-100 nm), which includes the mRNA vaccines and Alnylam's Patisiran siRNA, and any academic LNP or liposome work. However, I strongly disagree with Reviewer #1 that this kills the study for publication because determining the amount of escape from the endosome is an extremely difficult and crucial problem to address. However, this is a significant methodological deficiency that the authors need to ensure is thoroughly discussed in the text so that the non-specialist reader will appreciate the limitation of their study.

As described above we have made this limitation clearer in the discussion. The changes can be seen using markup in "Track changes" in the Word file.

And I also disagree with Reviewer #1 that this is not appropriate for a general audience. Quite the opposite, I think this study brings to light the complexities, deficiencies and unknowns of LNP delivery, something that most have likely physically experienced via the COVID mRNA vaccines.

Regarding Reviewer #1's concerns of hyperbole, I agree with the concern and the authors have satisfactorily toned down and tightened the text.

Overall, Hedlund et al. are quantitatively tackling a very difficult problem. They have bent over backwards to address all of the reviewers primary and secondary review concerns. Reviewer #1's concerns, though still quite negative, have resulted in a greatly improved manuscript. The study is important for both the specialist and general audience, and as such, the manuscript should be accepted.

Thank you for appreciating the work.